# One-Layer Transformer Provably Learns One-Nearest Neighbor In Context

**Zihao Li**[1*]   **Yuan Cao**[2*]   **Cheng Gao**[1]   **Yihan He**[1]   **Han Liu**[3]
**Jason M. Klusowski**[1]   **Jianqing Fan**[1†]   **Mengdi Wang**[1†]
[1]Princeton University   [2]The University of Hong Kong   [3]Northwestern University
{zihaoli,chenggao,yihan.he,jason.klusowski,jqfan,mengdiw}@princeton.edu
yuancao@hku.hk    hanliu@northwestern.edu

## Abstract

Transformers have achieved great success in recent years. Interestingly, transformers have shown particularly strong in-context learning capability – even without fine-tuning, they are still able to solve unseen tasks well purely based on task-specific prompts. In this paper, we study the capability of one-layer transformers in learning one of the most classical nonparametric estimators, the one-nearest neighbor prediction rule. Under a theoretical framework where the prompt contains a sequence of labeled training data and unlabeled test data, we show that, although the loss function is nonconvex when trained with gradient descent, a single softmax attention layer can successfully learn to behave like a one-nearest neighbor classifier. Our result gives a concrete example of how transformers can be trained to implement nonparametric machine learning algorithms, and sheds light on the role of softmax attention in transformer models.

## 1 Introduction

Transformers have emerged as one of the most powerful machine learning models since its introduction in Vaswani et al. [2017], achieving remarkable success in various tasks, including natural language processing [Devlin et al., 2018, Achiam et al., 2023, Touvron et al., 2023], computer vision [Dosovitskiy et al., 2020, He et al., 2022, Saharia et al., 2022], reinforcement learning [Chen et al., 2021, Janner et al., 2021, Parisotto et al., 2020], and so on. One intriguing aspect of transformers is their exceptional In-Context Learning (ICL) capability [Garg et al., 2022, Min et al., 2022, Wei et al., 2023, Von Oswald et al., 2023, Xie et al., 2021, Akyürek et al., 2022]. It has been observed that transformers can effectively solve unseen tasks solely relying on task-specific prompts, without the need for fine-tuning. However, the underlying mechanisms and reasons behind the exceptional in-context learning capability of transformers remain largely unexplored, leaving a significant gap in our understanding of how and why transformers can be pretrained to exhibit such remarkable performance.

Several recent studies have attempted to understand in-context learning (ICL) through the lens of learning specific function classes. Notably, Garg et al. [2022] proposed a well-defined approach: the training data includes a demonstration prompt, consisting of a sequence of labeled data and a new unlabeled query. The in-context learning performance of a transformer is then evaluated based on its ability to successfully execute a machine-learning algorithm to predict the query data label using the prompt demonstration (i.e., the context). Based on such definition, several works such as Zhang et al. [2023], Huang et al. [2023], Chen et al. [2024] investigated ICL the optimization dynamics

---

*Equal Contribution.

of transformers under in-context learning from a theoretical lens, but their studies are limited to linear regression prediction rules, which is a significant simplification of the transformer in-context learning task. Another line of work including Bai et al. [2024], Akyürek et al. [2022] investigated the expressiveness of transformers in context, but no optimization result is guaranteed. Whether transformers can handle more complicated ICL tasks under regular gradient-based training is still, in general, unknown.

In this paper, we examine the ability of single-layer transformers to learn the one-nearest neighbor prediction rule. Our major contributions are as follows:

- We establish convergence guarantees as well as prediction accuracy guarantees of a single-layer transformer in learning from examples of one-nearest neighbor classification. Utilizing the softmax attention layer, we demonstrate that the training loss can be minimized to zero despite the highly non-convex loss function landscapes. We further justify our results with numerical simulations.

- Based on the optimization results, we further establish a behavior guarantee for the trained transformer, demonstrating its ability to act like a 1-NN predictor under data distribution shift. Our result thus serves as a concrete example of how transformers can learn nonparametric methods, surpassing the scope of previous literature focusing on linear regression.

- In our technical analysis, we make the key observation that although the transformer loss is highly nonconvex when learning from one-nearest neighbor, its optimization process can be controlled by a two-dimensional dynamic system when choosing a proper initialization. By analyzing the behavior of such a system, we establish the convergence result despite the curse of nonconvexity.

To summarize, our result gives a concrete example of how transformers can be trained to implement nonparametric machine learning algorithms and sheds light on the role of softmax attention in transformer models. To our knowledge, this is the first paper that establishes a provable result in both optimization and consecutive behavior under distribution shift for a softmax attention layer beyond the scope of linear prediction tasks.

## 2 Preliminaries

In this section, we introduce the in-context learning data distribution based on the one-nearest neighbor data distribution and the setting of one-layer softmax attention transformers. Then, we discuss the training dynamics of transformers based on gradient descent.

### 2.1 In-Context Learning Framework: One-Nearest Neighbor

In an In-Context Learning (ICL) instance, the model is given a prompt $\{(\mathbf{x}_i, \mathbf{y}_i)\}_{i \in [N]} \sim \mathbb{P}_{\text{prompt}}$ and a query input $\mathbf{x}_{N+1} \sim \mathbb{P}_{\text{query}}$ from some data distributions $\mathbb{P}_{\text{prompt}}$ and $\mathbb{P}_{\text{query}}$, where $\{\mathbf{x}_i\}_{i \in [N]}$ are the input vectors, $\{\mathbf{y}_i\}_{i \in [N]} \subset \mathbb{R}$ are the corresponding labels (e.g. real-valued for regression, or $\{+1, -1\}$-valued for binary classification), and $\mathbf{x}_{N+1}$ is the query on which the model is required to make a prediction. Given a prompt $\{(\mathbf{x}_i, \mathbf{y}_i)\}_{i \in [N]}$, the prediction task is to predict an ground truth model $f(\mathbf{x}_{N+1}; \{(\mathbf{x}_i, \mathbf{y}_i)\}_{i \in [N]})$ that maps the query token $\mathbf{x}_{N+1}$ to a real number.

In this work, we consider using transformers as the model to perform in-context learning. For a prompt $\{(\mathbf{x}_i, \mathbf{y}_i)\}_{i \in [N]}$ of length $N$ and a query token $\mathbf{x}_{N+1}$, we consider use the following embedding:

$$\mathbf{H} = [\mathbf{h}_1, \mathbf{h}_2, \dots, \mathbf{h}_{N+1}] = \begin{bmatrix} \mathbf{x}_1 & \mathbf{x}_2 & \dots & \mathbf{x}_N & \mathbf{x}_{N+1} \\ \mathbf{y}_1 & \mathbf{y}_2 & \dots & \mathbf{y}_N & 0 \\ 0 & 0 & \dots & 0 & 1 \end{bmatrix} \in \mathbb{R}^{(d+2) \times (N+1)}. \quad (2.1)$$

We use the notation of $\mathbf{h}_j = [\mathbf{x}_j, \mathbf{y}_j, 0]$ for $j \leq N$, and $\mathbf{h}_{N+1} = [\mathbf{x}_{N+1}, 0, 1]$. Here, $\{\mathbf{x}_i\}_{i \in [N]}$ represents the input vectors, each associated with a corresponding label $\{\mathbf{y}_i\}_{i \in [N]}$, where $\mathbf{y}_i \in \mathbb{R}$ is the label. Throughout this paper, the sequence $\{(\mathbf{x}_i, \mathbf{y}_i)\}_{i \in [N]}$ are referred to as the *context* or *prompt* exchangeably. The $(d + 2)$-th row serves as the indicator for the training token, which equals to 0 value for $i \in [N]$ and 1 for $i = N + 1$, analogous to a positional embedding vector. Such an indicator allows the model to distinguish the query token from the context. Similar models have been

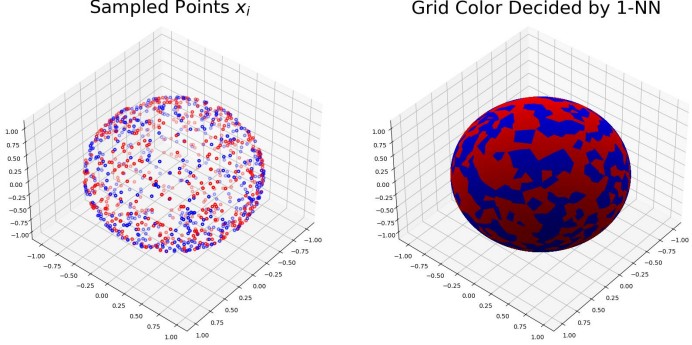

Figure 1: Illustration of data distribution in Assumption 1 on $\mathbb{S}^2$ and the corresponding ground-truth division of $\mathbb{S}^2$ generated by one-nearest neighbor. (1) In the left panel, the red and blue points correspond to the $\mathbf{x}_i$ with $\mathbf{y}_i = 1$ and $-1$ for $i \in [N]$, respectively, with $N = 500$. (2) In the right panel, the color of every point on the sphere is the same as its closest neighbor in $\{\mathbf{x}_i\}_{i \in [N]}$. The sphere is thus split into divisions by the one-nearest-neighbor decision rule.

studied in a line of recent works [Zhang et al., 2023, Huang et al., 2023, Chen et al., 2024, Bai et al., 2024, Akyürek et al., 2022] studying in-context learning of linear regression tasks.

Throughout this work, we focus on the case where the ground-truth prediction $f(\mathbf{x}_{N+1}; \{\mathbf{x}_i, \mathbf{y}_i\}_{i \in [N]})$ of the training data is constructed based on a One-Nearest Neighbor (1NN) data distribution, defined by the following definition.

**Definition 1** (One-Nearest Neighbor Predictor). *Given a prompt $\{(\mathbf{x}_i, \mathbf{y}_i)\}_{i \in [N]}$ and a query $\mathbf{x}_{N+1}$, we define the one-nearest neighbor predictor by*

$$\mathbf{y}_{i^*} := \sum_{i=1}^N \mathbb{1}(i = \underset{j \in [N]}{\operatorname{argmin}} \|\mathbf{x}_{N+1} - \mathbf{x}_j\|_2) \mathbf{y}_i.$$

*We also define $i^* = \operatorname{argmin}_{i \in [N]} \|\mathbf{x}_{N+1} - \mathbf{x}_i\|_2$.*

Without loss of generality, we assume that $\operatorname{argmin}_{j \in [N]} \|\mathbf{x}_{N+1} - \mathbf{x}_j\|_2$ is unique. Such assumption holds almost surely whenever $\{\mathbf{x}_i\}_{i \in [N]}$ is sampled from a continuous distribution. Notably, for a fixed prompt $\{(\mathbf{x}_i, \mathbf{y}_i)\}_{i \in [N]}$ and query $\mathbf{x}_{N+1}$, Definition 1 is identical to the nonparametric one-nearest neighbor estimator [Peterson, 2009, Beyer et al., 1999], in which the algorithm outputs the label corresponding to the vector closest to the input, with the prompt $\{(\mathbf{x}_i, \mathbf{y}_i)\}_{i \in [N]}$ as the training data in 1-NN.

Next, we discuss the distribution of the training dataset $\{(\mathbf{x}_i, \mathbf{y}_i)\} \cup \{\mathbf{x}_{N+1}\}$. Throughout the training process, we focus on the case in which $\{\mathbf{x}_i\}_{i \in [N+1]}$ are independently sampled from a uniform distribution on a $d-1$-dimensional sphere $\mathbb{S}^{d-1}$, $\{\mathbf{y}_i\}_{i \in [N]}$ is a zero-mean binary noise taken value in $\{+1, -1\}$, with $\{\mathbf{x}_i\}_{i \in [N+1]}$ and $\{\mathbf{y}_i\}_{i \in [N]}$ being independent. Our data distribution assumption can be summarized formally by the following assumption:

**Assumption 1** (Training Distribution). *For an embedding $\mathbf{H}$ defined by Eq. (2.1), we focus on the following underlying training distribution: (i) The sequence $\{\mathbf{x}_i\}_{i \in [N+1]}$ are sampled independently from a uniform distribution on a $d-1$ dimensional sphere $\mathbb{S}^{d-1} \subset \mathbb{R}^d$. (ii) The labels $\{\mathbf{y}_i\}_{i \in [N]}$ satisfies $\mathbb{E}[\mathbf{y}_i \mathbf{y}_j | \mathbf{x}_{1:N}] = 0$ and $\mathbb{E}[\mathbf{y}_i^2 | \mathbf{x}_{1:N}] = 1$ for all $i \neq j, i, j \in [N]$. (iii) We have $\mathbb{P}(\mathbf{y}_{1:N} | \mathbf{x}_{1:N}) = \mathbb{P}(\mathbf{y}_{1:N} | -\mathbf{x}_{1:N})$.*

Note that the case when $\{\mathbf{y}_i\}_{i \in [N]}$ and $\{\mathbf{x}_i\}_{i \in [N]}$ being independent when $\{\mathbf{x}_i\}_{i \in [N]}$, with $\{\mathbf{x}_i\}_{i \in [N]}$ are uniformly sampled from the sphere and $\{\mathbf{y}_i\}_{i \in [N]}$ are randomly sampled from $\{\pm 1\}$ is an example of Assumption 1. We remark that by considering the training data distribution in Assumption 1, we aim to study the capability of transformers in learning one-nearest neighbor prediction rules starting from the cleanest possible setting. Despite the seemingly simple problem setting, we would like to point out that this data distribution is still challenging to study, especially because of the assumption that the second order moment of $\{\mathbf{y}_i\}_{i \in [N]}$ and $\{\mathbf{x}_i\}_{i \in [N+1]}$ are uncorrelated. Due to

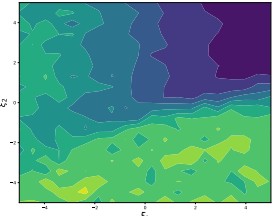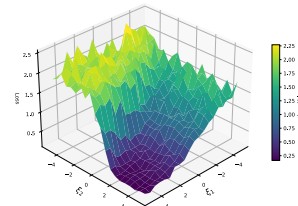

Figure 2: Heatmap and landscape of loss function of single layer transformer when learning from one-nearest neighbor. The loss is defined in Eq. (2.5), generated by sampling 100 training sequences according to Assumption 1, with $d = N = 4$. We parametrize $\mathbf{W}$ as $\mathrm{diag}\{\xi_1, \ldots, \xi_1, 0, \xi_2\}$.

such uncorrelation, the classifier given by one-nearest neighbor models are rather complicated. For example, Fig. 1 illustrates the randomly generated context data and the corresponding one-nearest neighbor prediction regions with $d = 3$ and $N = 500$, which clearly demonstrate the complexity of the training task. This further leads to a highly nonconvex and irregular objective function landscape, illustrated by Fig. 2.

## 2.2 One-Layer Softmax Attention Transformers

We consider a simplified version of the one-layer transformer architecture [Vaswani et al., 2017] that processes any input sequence $\mathbf{H}$ defined by Eq. (2.1) and outputs a scalar value:

$$\mathbf{H}_W = \mathbf{H} \cdot \mathrm{softmax}(\mathbf{H}^\top \mathbf{W}_K^\top \mathbf{W}_Q \mathbf{H}), \tag{2.2}$$

where $\mathrm{softmax}(\mathbf{A})$ applies softmax operator on each column of the matrix $\mathbf{A}$, i.e. $[\mathrm{softmax}(\mathbf{A})]_{ij} = \exp(A_{ij})/\sum_i \exp(A_{ij})$. Our model is slightly different from the standard self-attention transformers, as we consider a frozen value matrix. However, we also claim that such practice is common in deep learning theory Fang et al. [2020], Lu et al. [2020], Mei et al. [2018]. We also merge the query and key matrices into one matrix denoted as $\mathbf{W}$, which is often taken in recent theoretical frameworks [Zhang et al., 2023, Huang et al., 2023, Jelassi et al., 2022, Tian et al., 2023]. The output of the model is defined by the $(d+1)$-th element of the last column of $\mathbf{H}_{\mathbf{W}}$, with a closed form:

$$\widehat{\mathbf{y}}_{\mathbf{W}}(\mathbf{x}_{N+1}; \{\mathbf{x}_i, \mathbf{y}_i\}_{i \in [N]}) := [\mathbf{H}_{\mathbf{W}}]_{(d+1, N+1)} = \frac{\sum_{j=1}^N \mathbf{y}_j \exp(\mathbf{h}_j^\top \mathbf{W} \mathbf{h}_{N+1})}{\sum_{j=1}^{N+1} \exp(\mathbf{h}_j^\top \mathbf{W} \mathbf{h}_{N+1})}. \tag{2.3}$$

which is the weighted mean of $\mathbf{y}_1, \ldots, \mathbf{y}_N$. Here and after, we may occasionally suppress dependence on $\{\mathbf{x}_i, \mathbf{y}_i\}_{i \in [N]}$ and write $\widehat{\mathbf{y}}_{\mathbf{W}}(\mathbf{x}_{N+1}; \{\mathbf{x}_i, \mathbf{y}_i\}_{i \in [N]})$ as $\widehat{\mathbf{y}}_{\mathbf{W}}(\mathbf{x}_{N+1})$. Since the prediction takes only one entry of the token matrix output by the attention layer, actually only parts of $\mathbf{W}$ affect the prediction. To see this, we denote

$$\mathbf{W} = \begin{pmatrix} \mathbf{W}_{11} & \mathbf{W}_{12} & \mathbf{W}_{13} \\ \mathbf{W}_{21} & \mathbf{W}_{22} & \mathbf{W}_{23} \\ \mathbf{W}_{31} & \mathbf{W}_{32} & \mathbf{W}_{33} \end{pmatrix}, \tag{2.4}$$

with $\mathbf{W}_{11} \in \mathbb{R}^{d \times d}, \mathbf{W}_{21} \in \mathbb{R}^{1 \times d}, \mathbf{W}_{31} \in \mathbb{R}^{1 \times d}, \mathbf{W}_{12} \in \mathbb{R}^{d \times 1}, \mathbf{W}_{13} \in \mathbb{R}^{1 \times d}, \mathbf{W}_{22}, \mathbf{W}_{23}, \mathbf{W}_{32}$ and $\mathbf{W}_{33} \in \mathbb{R}$. Then by Eq. 2.3, it is easy to see that $\mathbf{W}_{i2}$ does not affect $\widehat{\mathbf{y}}_{\mathbf{W}}$ for $i \in [3]$, which means we can simply take all these entries as zero in the following sections. Notably, for a fixed prompt-query pair $\{(\mathbf{x}_i, \mathbf{y}_i)\}_{i \in [N]}$ and $\{\mathbf{x}_{N+1}\}$, such an architecture allows an arbitrarily close approximation to the 1-NN model: consider $\mathbf{W}_{11}^k = \xi_1^k I_d$ with $\xi_1^k$ goes to positive infinity, $\mathbf{W}_{33}^k = \xi_2^k$ such that $\xi_2^k - \xi_1^k$ converges to infinity, with the rest of $\mathbf{W}_{ij}^k$ bounded, then $\widehat{\mathbf{y}}_{\mathbf{W}^k}(\mathbf{x}_{N+1}))$ converges to $\mathbf{y}_{i^*}$ as $k$ goes to infinity.

## 2.3 Training Dynamics

To train the transformer model over the 1-NN task, we consider the Mean-Square Error (MSE) loss function. Specifically, the loss function is defined by

$$L(\mathbf{W}) = \frac{1}{2}\mathbb{E}_{\{\mathbf{x}_i,\mathbf{y}_i\}_{i\in[N]},\mathbf{x}_{N+1}}\left[\left(\widehat{\mathbf{y}}_{\mathbf{W}}(\mathbf{x}_{N+1}) - \mathbf{y}_{i^*}\right)^2\right]. \tag{2.5}$$

Above, the expectation is taken with respect to the sampled prompt $\{(\mathbf{x}_i,\mathbf{y}_i)\}_{i\in[N]}$ and the query $\mathbf{x}_{N+1}$. Notably, when the underlying distribution for the prompt and query are defined by Assumption 1, this loss function is nonconvex with respect to $\mathbf{W}$. Such nonconvexity makes the optimization hard to solve without further conditions such as PL condition or KL condition [Bierstone and Milman, 1988, Karimi et al., 2020]. We leave the proof of nonconvexity in Appendix E.

We shall consider the behavior of gradient descent on the single-layer attention architecture w.r.t. the loss function in Eq. (2.5). The parameters are updated as follows:

$$\mathbf{W}^{k+1} - \mathbf{W}^k = \frac{1}{\eta}\nabla_{\mathbf{W}}L(\mathbf{W}^k). \tag{2.6}$$

We shall consider the following initialization for the gradient descent:

**Assumption 2** (Initialization). *Let $\sigma > 0$ be a parameter. We assume the following initialization:*

$$\mathbf{W}^0 = \begin{pmatrix} 0_{(d+1)\times(d+1)} & 0_{d+1} \\ 0_{d+1} & -\sigma \end{pmatrix},$$

Here the parameter $\sigma$ is similar to masking, which is widely applied in self-attention training process, and prevents the model from focusing on the zero-label for the query $\mathbf{x}_{N+1}$, e.g. Vaswani et al. [2017], Baade et al. [2022], Chang et al. [2022]. The reason we take the zero initialization for non-diagonal entries will be made clear when we describe the proof in Section 4. However, from a higher view, it is because we want to keep the model focusing on the inner product between different $\mathbf{x}_i$, which largely reduces the complexity of the dynamic system under gradient descent and makes it tractable. We leave the question of convergence under alternative random initialization schemes for future work.

## 3 Main Results

In this section, we summarize the convergence of training loss and testing error respectively. In Section 3.1, we discuss the convergence of training loss under gradient descent. Specifically, we prove that with a proper initialization constant $\sigma$, gradient descent is able to minimize the loss function $L(\mathbf{W})$ despite the nonconvexity. In Section 3.2, we further discuss the testing error of the trained transformer under distribution shift. Specifically, we consider a distribution $\mathbb{P}_{\text{test}}$ for the prompt $\{(\mathbf{x}_i,\mathbf{y}_i)\}_{i\in[N]} \cup \{\mathbf{x}_{N+1}\}$, which is different from the training data distribution $\mathbb{P}_{\text{prompt}} \otimes \mathbb{P}_{\text{query}}$, and discuss the difference between the trained transformer and 1-NN predictor under such distribution shift.

### 3.1 Convergence of Gradient Descent

First, we prove that under suitable initialization parameter $\sigma$, the loss function will converge to zero under gradient descent.

**Theorem 1** (Convergence of Gradient Descent). *Consider performing gradient descent of the softmax-attention transformer model $\widehat{\mathbf{y}}_{\mathbf{W}}(\mathbf{x}_{N+1})$. Suppose the initialization satisfies Assumption 2 with $\sigma > 2(\max\{\log(Nd), -\log\left(1 - (N\sqrt{d})^{\frac{1}{d}}\right), C_d\left(1 - \frac{1}{2^N}\right)\})$, where $C_d = \text{poly}(d)$, and the number of context $N \geq O\left(\sqrt{d}\log d\right)$, then $L(\mathbf{W}^k)$ converges to 0.*

We leave the detailed proof in Appendix C. Theorem 1 shows that for the 1-NN data distribution, with a large enough initialization constant $\sigma$, the training loss of the transformer converges to zero under gradient descent. Here $\sigma$ plays a role similar to the masking techniques in the self-attention training training process, in which $\sigma$ is often set as infinity or an extremely large number. Such a technique has been widely accepted and shown to greatly accelerate the training process Vaswani et al. [2017], Devlin et al. [2018], Dosovitskiy et al. [2020]. We also compare our results to existing

works. Zhang et al. [2023] studied linear prediction tasks under gradient flow, however, their analysis is limited to linear attention layers. Huang et al. [2023] was the first to study softmax attention optimization under gradient descent, but their prediction is limited to linear prediction tasks under a finite orthogonal dictionary. Chen et al. [2024] established optimization convergence results for one-layer multi-head attention transformers under gradient flow. On the contrary, our work studies gradient descent convergence for transformer under a nonparametric estimator, setting it apart from all previous studies.

## 3.2 Results for New Task under Distribution shift

In this section, we discuss the behavior of trained transformers under distribution shifts, i.e., how the model *extrapolate* beyond the training distribution. Following the definition in Garg et al. [2022], let us assume in the training process, the prompts $\{(\mathbf{x}_i, \mathbf{y}_i)\}_{i \in [N]} \sim \mathbb{P}^{\text{train}}_{\text{prompt}}$, and the query $\mathbf{x}_{N+1} \sim \mathbb{P}^{\text{train}}_{\text{query}}$. During inference, the prompts and queries are sampled from a new distribution $\mathbb{P}^{\text{test}}$. We study the behavior of the trained transformers under possible prompt and query shift, i.e. $\mathbb{P}^{\text{test}} \neq \mathbb{P}^{\text{train}}_{\text{prompt}} \otimes \mathbb{P}^{\text{train}}_{\text{query}}$. Our studies show that, under some mild conditions, the behavior of the trained model is still similar to a 1-NN predictor even under a distribution shift. Before formally stating our result, let us introduce the following assumption on the testing distribution:

**Assumption 3** (Testing Distribution). *We make the following assumption on $\mathbb{P}^{\text{test}}$:*

 (i) *There exists a $R \geq 0$ such that $|\mathbf{y}_i| \leq R$ holds for all $\mathbf{y}_i$ sampled from $\mathbb{P}^{\text{test}}$.*

 (ii) *For all $\{(\mathbf{x}_i, \mathbf{y}_i)\}_{i \in [N]} \cup \{\mathbf{x}_{N+1}\} \sim \mathbb{P}^{\text{test}}$, we have $\mathbf{x}_i \in \mathbb{S}^{d-1}$ for all $i \in [N+1]$.*

Note that Assumption 3 only requires the label $\mathbf{y}_i$ is bounded and $\mathbf{x}_i$ is supported on a sphere. We also remind the reader that we do not assume independence between different $\mathbf{x}_i$ or $\{\mathbf{x}_i\}_{i \in [N+1]}$ and $\{\mathbf{y}_i\}_{i \in [N+1]}$. Now we are ready to summarize our result in the following theorem.

**Theorem 2** (Resemblance to 1-NN predictor under Distribution Shift). *Suppose Assumption 1 and 3 hold for $\mathbb{P}^{\text{train}}_{\text{prompt}} \otimes \mathbb{P}^{\text{train}}_{\text{query}}$ and $\mathbb{P}^{\text{test}}$. If we define*

$$A_\delta := \{\|\mathbf{x}_j - \mathbf{x}_{N+1}\|_2^2 \geq \|\mathbf{x}_{i^*} - \mathbf{x}_{N+1}\|_2^2 + \delta \text{ for all } j \neq i^* \text{ such that } \mathbf{y}_j \neq \mathbf{y}_{i^*}\},$$

*then, after $K$-iterations of gradient descent, we have*

$$\mathbb{E}_{\{(\mathbf{x}_i, \mathbf{y}_i)\}_{i \in [N]}, \mathbf{x}_{N+1}}\left[\left(\widehat{\mathbf{y}}_{\mathbf{W}^K}(\mathbf{x}_{N+1}) - \mathbf{y}_{i^*}\right)^2\right] \leq O\left(\inf_\delta \left\{R^2 N^2 K^{-\text{poly}(N,d)\delta} + R^2 \mathbb{P}^{\text{test}}(A_\delta^c)\right\}\right),$$

*here the expectation is taken w.r.t $\{(\mathbf{x}_i, \mathbf{y}_i)\}_{i \in [N]} \cup \{\mathbf{x}_{N+1}\} \sim \mathbb{P}^{\text{test}}$. Recall that $\mathbf{y}_{i^*}$ is the 1-NN predictor of $\mathbf{x}_{N+1}$, which we defined in Definition 1.*

We leave the detailed proof in Appendix D. Let us discuss the implication of Theorem 2. The event $A_\delta$ describes the situation when the query $\mathbf{x}_{N+1}$ is located at an "inner point" away from its decision boundary, in which its distance to the nearest neighbor $\mathbf{x}_{i^*}$ is strictly larger than all other points. Such a quantity is similar to the margin condition in classification theory in deep learning Bartlett et al. [2017] and $k$-NN literature Chaudhuri and Dasgupta [2014], where the optimal choice probability is strictly larger than all suboptimal choices. Specifically, if $\mathbb{P}^{\text{test}}(A_{\delta^*}) = 1$ for some $\delta^* > 0$, i.e., the query $\mathbf{x}_{N+1}$ is strictly bounded away from the decision boundary almost surely, then the $L_2$ distance between $\widehat{\mathbf{y}}_{\mathbf{W}^k}$ and the 1-NN predictor will converge in a $O(R^2 K^{-\text{poly}(N,d)\delta^*})$ even under a shifted distribution. We also introduce the following corollary, in which we show that when $\mathbf{y}_i$ only takes value in a finite integer set, resembling a classification task, the trained transformer behaves like a 1-NN predictor under an additional rounding operation.

**Corollary 1** (Classfication of Trained Transformer). *Suppose $\mathbf{y}_i \in [M]$ for some integer $M \geq 0$ under $\mathbb{P}^{\text{test}}$, then we have*

$$\mathbb{P}_{test}\left(\text{Round}\left(\widehat{\mathbf{y}}_{\mathbf{W}^k}(\mathbf{x}_{N+1})\right) \neq \mathbf{y}_{i^*}\right) \leq O\left(\inf_\delta \left\{M^2 N^2 K^{-\text{poly}(N,d)\delta} + M^2 \mathbb{P}^{\text{test}}(A_\delta^c)\right\}\right).$$

*Here we define*

$$\text{Round}(t) := \mathbb{1}_{[t] < \frac{1}{2}} \lfloor t \rfloor + \mathbb{1}_{[t] \geq \frac{1}{2}} \lceil t \rceil,$$

*i.e. the mapping from $t \in \mathbb{R}$ to its closest integer, and $A_\delta$ is defined as in Theorem 2. Moreover, if there exists $\delta^* > 0$ such that $\mathbb{P}^{test}(A_{\delta^*}) = 0$, then we have*

$$\mathbb{P}^{test}\big( \text{Round}\,\big(\widehat{\mathbf{y}}_{\mathbf{W}^k}(\mathbf{x}_{N+1})\big) \neq \mathbf{y}_{i^*}\big) = 0$$

*whenever $K \geq O\big(\frac{\log(MN)}{\text{poly}(N,d)\delta^*}\big)$.*

We leave the detailed proof in Appendix D. Corollary 1 provides a convergence rate for the classification difference between 1-NN and the pretrained transformer. Notably, when $\mathbf{x}_{N+1}$ is well separated from the decision boundary in the testing distribution $\mathbb{P}_{\text{test}}$, the trained transformer will behave exactly the same as the 1-NN classifier in $O\big(\frac{\log(MN)}{\text{poly}(N,d)\delta^*}\big)$ gradient steps for the pretrained transformer. Theorem 2 and Corollary 1 show that the trained transformer under gradient descent is robust to both query and prompt distribution shift in the test distribution $\mathbb{P}^{ptest}$, in the sense that it will maintain its resemblance to a 1-NN predictor in both prediction and classification task, thus extended the results in Zhang et al. [2023], Huang et al. [2023], Chen et al. [2024] to a nonparametric estimator.

# 4 Sketch of Proof

In this section, we sketch the proof of Theorem 1 and highlight the techniques we used. The full proof is left to Appendix C.

**Equivalence to a Two-Dimensional Dynamic System.** Recall that $\{\mathbf{x}_i\}_{i \in [N+1]}$ and the first and second moment of $\{\mathbf{y}_i\}_{i \in [N]}$ are uncorrelated. Utilizing this uncorrelation between $\{\mathbf{x}_i\}_{i \in [N+1]}$ and $\{\mathbf{y}_i\}_{i \in [N]}$, we can eliminate the reliance of the gradient on $\{\mathbf{y}_i\}_{i \in [N]}$ since we are considering a population loss. Moreover, utilizing the structure of the initialization, we can prove by induction that all $\mathbf{W}_{ij}$ will remain zero except for $\mathbf{W}_{11}$ and $\mathbf{W}_{33}$. This shows that with a suitable initialization, the transformer model will only focus on the relationship between different tokens $\mathbf{x}_i$ throughout the whole training process. Our findings can be summarized by the following lemma.

**Lemma 1** (Closed-Form Gradient). *With the initialization in Assumption 2, the gradient of $L(\mathbf{W}^k)$ with respect to $\mathbf{W}_{11}$ can be written in the following form for all $k \geq 0$:*

$$\nabla_{\mathbf{W}_{11}} L(\mathbf{W}^k) = \mathbb{E}\left[ \sum_{i=1}^{N} g_i^k(\mathbf{x}_i^\top \mathbf{x}_{N+1}) \cdot \mathbf{x}_i \mathbf{x}_{N+1}^\top + g_{i^*}^k(\mathbf{x}_{i^*}^\top \mathbf{x}_{N+1}) \cdot \mathbf{x}_{i^*} \mathbf{x}_{N+1}^\top \right] \qquad (4.1)$$

*where $\{g_i^k(x)\}_{i \in [N]} \cup \{g_{i^*}^k(x)\} : \mathbb{R} \to \mathbb{R}$ is a set of functions. Here the expectation is taken with respect to $\{\mathbf{x}_i\}_{i \in [N+1]}$, with $\mathbf{x}_{i^*} = \arg\min_{\mathbf{x} \in \{\mathbf{x}_i\}_{i \in [N]}} \|\mathbf{x} - \mathbf{x}_{N+1}\|_2$ sampled i.i.d. from a uniform distribution on $\mathbb{S}^{d-1}$. Moreover, we have $\nabla_{\mathbf{W}_{ij}} L(\mathbf{W}^k) = 0$ for all $(i,j) \in [3] \times [3]$ and all $k \geq 0$ except for $\mathbf{W}_{11}$ and $\mathbf{W}_{33}$.*

Lemma 1 shows that we only need to consider $\mathbf{W}_{11}$ and $\mathbf{W}_{33}$ in our update since all other entries will remain zero during the whole learning process. Note that in Eq. (4.1), all nonlinearity comes from the inner product between $\mathbf{x}_i^\top \mathbf{x}_{N+1}$ and $\mathbf{x}_{i^*}^\top \mathbf{x}_{N+1}$. Recall that $\{\mathbf{x}_i\}_{i \in [N+1]}$ are i.i.d. sampled from a uniform distribution supported on a $d-1$-dimensional sphere $\mathbb{S}^{d-1}$, therefore, the distribution of $\{\mathbf{x}_i\}_{i \in [N+1]}$ is rotational invariance, which means $\bigotimes_{i \in [N+1]} \mathrm{P}_{\mathbf{x}_i} = \bigotimes_{i \in [N+1]} \mathrm{P}_{U\mathbf{x}_i}$ for all orthogonal matrix $U \in \mathbb{R}^{d \times d}$. Since the rotation of $\{\mathbf{x}_i\}_{i \in [N+1]}$ does not change the inner products $\{\mathbf{x}_i^\top \mathbf{x}_i\}_{i \in [N]}$ and $\mathbf{x}_{i^*}^\top \mathbf{x}_{N+1}$, from the structure of $\nabla_{\mathbf{W}_{11}} L(\mathbf{W}^k)$ illustrated by Eq. (4.1), we shall always have $U \nabla_{\mathbf{W}_{11}} L(\mathbf{W}^k) U^\top = \nabla_{\mathbf{W}_{11}} L(\mathbf{W}^k)$, which shows $\nabla_{\mathbf{W}_{11}} L(\mathbf{W}^k) = c_k I_d$ for some constant $c_k$ by simple algebra. We summarize our result in the following lemma.

**Lemma 2** (Two-Dimensional System). *With the initialization in Assumption 2, there exists two sets of real numbers $\{\xi_1^k\}_{k \geq 0}$ and $\{\xi_2^k\}_{k \geq 0}$, such that $\mathbf{W}^k$ has the following form:*

$$\mathbf{W}^k = \text{diag}\{\underbrace{\xi_1^k, \ldots, \xi_1^k}_{d \text{ times}}, 0, -\xi_2^k\}.$$

With Lemma 2, we reduce the dimension of the original dynamic system in Eq. (2.6) from $(d+2)^2$ to 2. We now only need to focus on the evolution of $\xi_1^k$ and $\xi_2^k$ for all $k \geq 0$.

**Convergence of the Dynamic System.** Lemma 2 helps us largely reduce the dimension of the training dynamics. However, this does not make our question a trivial one, as the loss function is still highly nonconvex even when we only need to consider a two-dimensional subspace of $\mathbb{R}^{(d+2)\times(d+2)}$. To see this, we introduce the following lemma:

**Lemma 3** (Nonconvexity of Transformer Optimization). *When $\mathbf{W}$ lie in a two-dimensional subspace of $\mathbb{R}^{(d+2)\times(d+2)}$ defined by $\mathbf{W} = \mathrm{diag}\{\underbrace{\xi_1, \ldots, \xi_1}_{d \text{ times}}, 0, -\xi_2\}$, the original loss function defined in Eq. (2.5) is equivalent to the following:*

$$L(\xi_1, \xi_2) := \mathbb{E}\left[\left(\frac{\sum_{j=1}^N \exp(\xi_1 \langle \mathbf{x}_j, \mathbf{x}_{N+1}\rangle)\mathbf{y}_j}{\sum_{i=1}^N \exp(\xi_1 \langle \mathbf{x}_i, \mathbf{x}_{N+1}\rangle) + \exp(\xi_1 - \xi_2)} - \mathbf{y}_{i^*}\right)^2\right]. \quad (4.2)$$

*Such loss function is still nonconvex.*

We leave the detailed proof in Appendix E. Nonconexity shown by Lemma 3 implies attaining the global minimum could be hard. Previous works such as Zhang et al. [2023] utilize conditions such as Polyak-Lojasiewicz inequality to analyze such systems, however, those conditions are not applicable in our setting, and a more delicate analysis for the evolution of $\xi_1^k$ and $\xi_2^k$ is needed. We characterize their behavior by the following lemma.

**Lemma 4.** *For $\xi_1^k \geq 0$, there exists constants $c_1, c_2, c_3, c_4 > 0$, such that*

$$\frac{d}{\eta}(\xi_1^{k+1} - \xi_1^k) \geq c_1 \cdot \exp(-6\xi_1^k) - c_2 \cdot \exp(2\xi_1^k - \xi_2^k),$$

*and*

$$\frac{d}{\eta}(\xi_1^{k+1} - \xi_1^k) \leq c_3 \cdot \exp\left(\mathrm{poly}(N, d) \cdot \xi_1^k\right) - c_4 \cdot \exp\left(2(\xi_1^k - \xi_2^k)\right)$$

Lemma 4 shows that there exits a constant $c_b \in (0, 1)$, such that $\xi_1^k$ will keep increasing with a scale of $\Omega(\eta \log k)$ until $\xi_1^k \leq c_b \xi_2^k$. With this ratio, we obtain the following lemma for the increment of $\xi_2^k$.

**Lemma 5.** *For $\xi_1^k \geq 0$, there exits constant $c_1', c_2'$, such that*

$$c_1' \cdot \exp(-\mathrm{poly}(N, d) \cdot \xi_2^k) \leq \frac{1}{\eta}(\xi_2^{k+1} - \xi_2^k) \leq c_2' \cdot \exp(-\mathrm{poly}(N, d) \cdot \xi_2^k).$$

Lemma 5 shows that $\xi_2^k$ will monotonically increase with a scale of $\Omega(\eta \exp(\xi_2^k))$, which implies $\xi_2^k = \Omega(\log k)$. Combining Lemma 4 and 5, we show that both $\xi_1^k$ and $\xi_2^k$ converge to infinity, with $\xi_1^k$ maintaining a slower speed, as its decreases when getting closer to $\xi_2^k$ from the below. Recall that the loss function is equivalent to Eq. (4.2) under the initialization specified in Assumption 2, which shows that $L(\xi_1^k, \xi_2^k)$ will converge to zero as long as $\xi_1^k$ and $\xi_2^k - \xi_1^k$ both converges to infinity. We thus conclude our proof of $L(\mathbf{W}^k)$ eventually converges to it global minimum.

# 5 Numerical Results

In previous sections, we have shown that with the initialization specified in Assumption 2, a single softmax attention layer transformer is able to learn the 1-NN predictor under gradient descent and remain robust under distribution shift. We now conduct experiments in a less restrictive setting and show that even without specific initialization and full-batch gradient descent, simple stochastic gradient descent updates with random parameter initialization for the parameters are still sufficient for the model to learn the 1-NN predictor. First, we investigate the convergence of single-head single-layer transformers [Vaswani et al., 2017] trained on 1-NN tasks. The training data are sampled from $\mathbb{P}_{\text{prompt}}^{\text{train}}$ and $\mathbb{P}_{\text{query}}^{\text{train}}$, defined in Assumption 1. We choose context length $N \in \{16, 32, 64\}$ and input dimension $d \in \{8, 16\}$. The model is trained on a dataset with a size of 10000, and an epoch

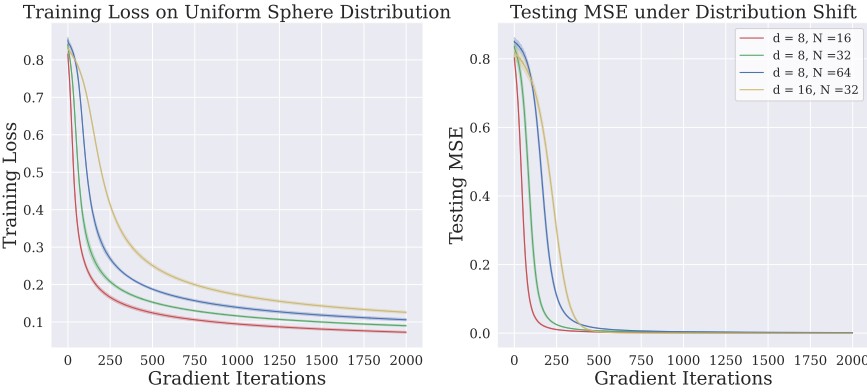

Figure 3: Prediction error for single softmax attention layer as a function of gradient iteration number. (1) The left panel shows the convergence of loss function during the training process. (2) The right panel shows the MSE between the trained model and a 1-NN predictor on a well-separated testing dataset under distribution shift, as we discuss in Section 5. Curves and error bars in both panels are computed as twice the standard deviation based on 10 independent trials.

number of 2000. To ensure our training convergence result holds beyond the gradient descent scheme, we choose SGD as our optimizer, with a batch size of 128 and a learning rate of 0.1. We use the random Gaussian as our initialization. Our results for the training loss convergence are summarized in the left panel of Fig. 3. The results show that the model converges to 1-NN predictor on the training data even under SGD and random initialization. Moreover, as the dimension $d$ and the length of contexts $N$ becomes larger, the convergence speed becomes slower.

To verify our results on the distribution shift, we generate testing data sampled from a distribution difference from the training data, and report the mean square error between the model prediction and the 1-NN predictor. Furthermore, the testing data satisfies $\mathbb{P}(A_\delta^*) = 1$, with $\delta^*$ specified as 0.1. Recall that we defined $A_\delta$ in Theorem 2 as the event where $\mathbf{x}_{N+1}$ is separated from the decision boundary with a distance of at least $\delta$. We leave the details of our data-generating process in Appendix B. We test the trained transformer model on this dataset once every epoch throughout the training process. Our results are summarized in the right panel of Fig. 3. The results show that the testing error decreases much faster than the training loss, due to the boundary separation condition, which the uniformly-sampled training data do not enjoy. Our result also coincides with our theoretical result in Theorem 2, showing that the trained transformers are robust under distribution shift, and benefits greatly from staying away from the decision boundary.

## 6    Conclusion

We investigate the ability of single-layer transformers to learn the one-nearest neighbor prediction rule, a classic nonparametric estimator. Under a theoretical framework where the prompt contains a sequence of labeled training data and unlabeled test data, we demonstrate that, despite the non-convexity of the loss function during gradient descent training, a single softmax attention layer can successfully emulate a one-nearest neighbor predictor. We further show that the trained transformer is robust to the distribution shift of the testing data. As far as we know, this paper is the first to establish training convergence and behavior under distribution shifts for softmax attention transformers beyond the domain of linear predictors.

## Acknowledgments and Disclosure of Funding

We thank the anonymous reviewers for their helpful comments. Yuan Cao is partially supported by NSFC 12301657 and Hong Kong RGC-ECS 27308624. Han Liu's research is partially supported by the NIH R01LM01372201. Jason M. Klusowski was supported in part by the National Science

Foundation through CAREER DMS-2239448, DMS-2054808 and HDR TRIPODS CCF-1934924. Jianqing Fan's research was partially supported by NSF grants DMS-2210833, DMS-2053832, and ONR grant N00014-22-1-2340. Mengdi Wang acknowledges the support by NSF IIS-2107304, NSF CPS-2312093, ONR 1006977 and Genmab.

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

## A  Related Works

**In-Context Learning.**  In-context learning refers to transformers' ability to solve unseen tasks without fine-tuning. Min et al. [2022] studied which aspects of the demonstrations contribute to end-task performance. Further, Garg et al. [2022] empirically investigated the ability of transformer architectures to learn a variety of function classes in context. From a theoretical perspective, Bai et al. [2024], Abernethy et al. [2024], Akyürek et al. [2022] studied the expressiveness of transformers to approximate statistical algorithms. Li et al. [2023a] studied the generation ability of transformers in ICL tasks. Jeon et al. [2024] studies the information-theoretical lower bound of in-context learning. However, none of these works studied the optimization process when training a transformer in context.

**Optimization of Transformers.**  There are various works that studied optimization for transformers. Among all these studies, Huang et al. [2023], Zhang et al. [2023], Chen et al. [2024] studied the optimization dynamics of transformers of learning linear prediction tasks in context. Specifically, Zhang et al., 2023 studied the gradient flow dynamics of linear self-attention model and obtained convergence results utilizing PL condition. Huang et al. [2023] studied linear prediction tasks with a softmax-attention layer trained with gradient descent, with a finite support prompt/query distribution. Chen et al. [2024] studied the gradient flow dynamics in training multi-head softmax attention on linear prediction tasks. Ahn et al. [2024], Giannou et al. [2024] studied transformers' ability to learn optimization methods. Prior works also studied transformer optimization beyond in-context learning tasks. Tian et al. [2023] studied a single-layer transformer with one self-attention layer plus one decoder layer, and proved that the attention layer acts as a scanning algorithm. Li et al. [2023b] studied the optimization of transformers in learning semantic structures. Jelassi et al. [2022] studies the spatial localization property of vision transformer in optimization.

**Notations.**  We write $[N] = \{1, \ldots, N\}$. For two vector $\mathbf{x}^1 = (x_1^1, \ldots, x_d^1)$ and $\mathbf{x}^2 = (x_1^2, \ldots, x_d^2)$, we write their inner product $\sum_{i=1}^{d} x_i^1 x_i^2$ as $\mathbf{x}^{1\top}\mathbf{x}^2$. We use $0_n$ and $0_{m \times n}$ to denote the zero vector and zero matrix of size $n$ and $m \times n$, respectively. We write $\{\mathbf{x} \in \mathbb{R}^d : \|\mathbf{x}\|_2 = 1\}$ as $\mathbb{S}^{d-1}$. For two series $\{a_k\}_{k \geq 0}$ and $\{b_k\}_{k \geq 0}$, we write $a_k = \Omega(b_k)$ if there exists $0 < C_1, C_2$ such that $C_1 \cdot b_k \leq a_k \leq C_2 \cdot b_k$. We write $a_k = O(b_k)$ if there exists $C > 0$ such that $a_k \leq C \cdot b_k$. We use $a_k = \mathrm{poly}(b_k)$ if there exits an $n$-degree polynomial $P_n(x)$ such that $a_k = O(P_n(b_k))$. For $(a_k)_{k \in [N]}$, we define its permutation $(a_{(k)})_{k \in [N]}$ such that $a_{(1)} \geq a_{(2)} \geq \ldots \geq a_{(n)}$. We use $I_d$ to denote the $d$-dimensional identity matrix and sometimes we also use $I$ when the dimension is clear from the context. Unless otherwise defined, we use lower case letters for scalars and vectors and use upper case letters for matrices. For a matrix $\mathbf{A}$ we denote its entry in the $i$-th row and $j$-th column by $[\mathbf{A}]_{ij}$.

## B  Data-Generating Process in Experiment

In our experiment, we generate the testing dataset with context length $N$ and dimension $d$ with separation parameter $\delta^*$ such that $\|\mathbf{x}_j - \mathbf{x}_{N+1}\|_2^2 \geq \|\mathbf{x}_{i^*} - \mathbf{x}_{N+1}\|_2^2 + \delta$ for all $j \in [N], j \neq i^*$ by the following procedure:

   (i) We sample $\mathbf{x}_i$ from the uniform distribution on $\mathbb{S}^{d-1}$ and $\mathbf{y}_i \sim \mathcal{N}(0, 1)$ for all $i \in [N]$;

  (ii) We random sample $i^* \in [N]$ by uniform distribution and set $\mathbf{x}_{N+1} = \mathbf{x}_{i^*}$ with the 1-NN label being $\mathbf{y}_{i^*}$;

 (iii) If $\|\mathbf{x}_j - \mathbf{x}_{N+1}\|_2^2 \leq \delta$, set $\mathbf{x}_j = -\mathbf{x}_j$.

## C  Proof for Theorem 1

In this section, we elaborate on the proof of Theorem 1. Our proof can be broken down into the following steps:

   (i) With induction, we prove that the evolution dynamics of $\mathbf{W}^k$ under gradient descent can be captured by a two-parameter dynamic system, parametrized by $\xi_1^k$ and $\xi_2^k$;

(ii) By estimating the update dynamics for $\xi_1^k$ and $\xi_2^k$, we prove that with a proper initialization parameter $\sigma$, We will have $\xi_1^k, \xi_2^k = \Omega(\log k)$, with $\xi_1^k \leq c \cdot \xi_2^k$ for a constant $c \in (0, 1)$.

(iii) With the non-asymptotic behavior of $\xi_1$ and $\xi_2$ determined, we further control the loss function $L(\mathbf{W}^k)$ and establish a convergence for the loss function.

In the following sections, we will discuss how we prove those three items in Section C.1, C.2 and C.3, respectively.

## C.1 Dynamic of Gradient Descent

In this section, we prove that we can characterize the evolution of $\mathbf{W}$ under gradient descent with a two-parameter dynamic system. We proceed by mathematical induction:

(1) We prove that when $\mathbf{W}^k$ is a diagonal matrix with $[\mathbf{W}^k]_{[d]\times[d]} = c_k I_d$ for some constant $c_k$, we can have the following break downs of our proof:

$$\frac{1}{\eta}(\mathbf{W}^{k+1} - \mathbf{W}^k) = \mathrm{diag}\{\underbrace{\Delta\xi_1^0, \ldots, \Delta\xi_1^0}_{d \text{ times}}, 0, -\Delta\xi_2^0\}, \tag{C.1}$$

where $\Delta\xi_1^0$ and $\Delta\xi_2^0$ are two positive constants.

(2) By $\mathbf{W}^k$ being a diagonal matrix with $[\mathbf{W}^k]_{[d]\times[d]} = c_k I_d$, combined with Eq. (C.1), we prove that $\mathbf{W}^{k+1}$ is a diagonal matrix and $[\mathbf{W}^{k+1}]_{[d]\times[d]} = c_{k+1}I_d$. .

Now since (1) is naturally satisfied by the initialization in Assumption 2 for $k = 0$, we can conclude the proof by simply proving (1) for $k \geq 1$.

**Proof for Step (1) of Induction.** Recall that our loss function can be written as

$$L(\mathbf{W}) = \frac{1}{2}\mathbb{E}_{\{\mathbf{x}_i,\mathbf{y}_i\}_{i\in[N]};\mathbf{x}_{N+1}}\left[\left(\hat{\mathbf{y}}_{\mathbf{W}}(\mathbf{x}_{N+1}) - f(\mathbf{x}_{N+1};\{\mathbf{x}_i,\mathbf{y}_i\}_{i\in[N]})\right)^2\right]$$

$$= \frac{1}{2}\mathbb{E}_{\{\mathbf{x}_i,\mathbf{y}_i\}_{i\in[N]};\mathbf{x}_{N+1}}\left[\left(\frac{\sum_{j=1}^N \exp\left(\mathbf{x}_j^\top \mathbf{W}_{11}\mathbf{x}_{N+1} + \mathbf{y}_j^\dagger \mathbf{W}_j^\ddagger \mathbf{x}_{N+1} + \mathbf{x}_j\top\mathbf{W}_{13} + \mathbf{y}_j^\dagger \mathbf{W}_j^\dagger\right)\mathbf{y}_j}{\sum_{j=1}^{N+1} \exp\left(\mathbf{x}_j^\top \mathbf{W}_{11}\mathbf{x}_{N+1} + \mathbf{y}_j^\dagger \mathbf{W}_j^\ddagger \mathbf{x}_{N+1} + \mathbf{x}_j\top\mathbf{W}_{13} + \mathbf{y}_j^\dagger \mathbf{W}_j^\dagger\right)} - \mathbf{y}_{i^*}\right)^2\right]$$

where

$$\mathbf{W} = \begin{pmatrix} \mathbf{W}_{11} & \mathbf{W}_{12} & \mathbf{W}_{13} \\ \mathbf{W}_{21} & \mathbf{W}_{22} & \mathbf{W}_{23} \\ \mathbf{W}_{31} & \mathbf{W}_{32} & \mathbf{W}_{33} \end{pmatrix},$$

with $\mathbf{W}_{11} \in \mathbb{R}^{d\times d}, \mathbf{W}_{21} \in \mathbb{R}^{1\times d}, \mathbf{W}_{31} \in \mathbb{R}^{1\times d}, \mathbf{W}_{12} \in \mathbb{R}^{d\times 1}, \mathbf{W}_{13} \in \mathbb{R}^{1\times d}, \mathbf{W}_{22}, \mathbf{W}_{23}, \mathbf{W}_{32}$ and $\mathbf{W}_{33} \in \mathbb{R}$, and we make the additional definition of $\mathbf{y}_j^\dagger = \mathbf{y}_j, \mathbf{W}_j^\dagger = \mathbf{W}_{23}, \mathbf{W}_j^\ddagger = \mathbf{W}_{21}$ for $j \in [N]$ and $\mathbf{y}_{N+1}^\dagger = 1, \mathbf{W}_{N+1}^\dagger = \mathbf{W}_{33}, \mathbf{W}_{N+1}^\ddagger = \mathbf{W}_{31}$. Throughout the rest of this paper, we will also adopt the notation of

$$\mathbf{q}_j(\mathbf{x}, \mathbf{W}) := \frac{\exp\left(\mathbf{x}_j^\top \mathbf{W}_{11}\mathbf{x}_{N+1} + \mathbf{y}_j^\dagger \mathbf{W}_j^\ddagger \mathbf{x}_{N+1} + \mathbf{x}_j\top\mathbf{W}_{13} + \mathbf{y}_j^\dagger \mathbf{W}_j^\dagger\right)}{\sum_i \exp\left(\mathbf{x}_i^\top \mathbf{W}_{11}\mathbf{x}_{N+1} + \mathbf{y}_i^\dagger \mathbf{W}_j^\ddagger \mathbf{x}_{N+1} + \mathbf{x}_i\top\mathbf{W}_{13} + \mathbf{y}_j^\dagger \mathbf{W}_j^\dagger\right)} \tag{C.2}$$

when there is no ambiguity, where $q : \mathbf{W} \times \{\mathbf{x}_i\}_{i\in[N+1]} \times \{\mathbf{y}_i\}_{i\in[N]}$. As we have discussed in Section 2.2, $\mathbf{W}_{*2}$ does not affect the outcome of the transformer, therefore the second column will remain 0 throughout the training procedure. Now we will calculate the gradient $\nabla_{\mathbf{W}_{ij}} L(\mathbf{W}^0)$ respectively. Note that for $\mathbf{f}_\theta^j(x) = \exp(\mathbf{g}_\theta^j(x))$, we have

$$\nabla_\theta\left(\frac{\sum_j \mathbf{f}_\theta^j(x)\cdot\mathbf{y}_j}{\sum_j \mathbf{f}_\theta^j(x)} - \mathbf{y}_{i^*}\right)^2 = 2\left(\frac{\sum_j \mathbf{f}_\theta^j(x)\cdot\mathbf{y}_j}{\sum_j \mathbf{f}_\theta^j(x)} - \mathbf{y}_{i^*}\right)\cdot\left\{\frac{\sum_j \mathbf{f}_\theta^j(x)\mathbf{y}_j\nabla_\theta\mathbf{g}_\theta^j}{\sum_j \mathbf{f}_\theta^j(x)} - \frac{\sum_j \mathbf{f}_\theta^j(x)\nabla_\theta\mathbf{g}_\theta^j(x)}{\sum_{j=1}^{N+1}\mathbf{f}_\theta^j(x)}\cdot\frac{\sum_j \mathbf{f}_\theta^j(x)\mathbf{y}_j}{\sum_j \mathbf{f}_\theta^j(x)}\right\},$$

therefore, with some algebra, we have the following closed-form formula for $\nabla_{\mathbf{W}_{ij}} L(\mathbf{W})$ respectively,

$$
\nabla_{\mathbf{W}_{11}} L(\mathbf{W}) = \mathbb{E}\Bigg[ \Big\{ \sum_{j=1}^{N+1} \mathbf{q}_j(\mathbf{x},\mathbf{W})(\mathbf{x}_j \mathbf{x}_{N+1}^\top)\mathbf{y}_j - \big\{ \sum_{j=1}^{N+1} \mathbf{q}_j(\mathbf{x},\mathbf{W})(\mathbf{x}_j \mathbf{x}_{N+1}^\top) \big\} \big\{ \sum_{j=1}^{N+1} \mathbf{q}_j(\mathbf{x},\mathbf{W})\mathbf{y}_j \big\} \Big\}
$$
$$
\cdot \Big\{ \sum_{j=1}^{N+1} \mathbf{q}_j(\mathbf{x},\mathbf{W})(\mathbf{y}_j - \mathbf{y}_{i^*}) \Big\} \Bigg], \tag{C.3}
$$

$$
\nabla_{\mathbf{W}_{21}} L(\mathbf{W}) = \mathbb{E}\Bigg[ \Big\{ \sum_{j=1}^{N} \mathbf{q}_j(\mathbf{x},\mathbf{W})(\mathbf{y}_j \mathbf{x}_{N+1}^\top)(\mathbf{y}_j) - \big\{ \sum_{j=1}^{N} \mathbf{q}_j(\mathbf{x},\mathbf{W})(\mathbf{y}_j \mathbf{x}_{N+1}^\top) \big\} \big\{ \sum_{j=1}^{N+1} \mathbf{q}_j(\mathbf{x},\mathbf{W})(\mathbf{y}_j) \big\} \Big\}
$$
$$
\cdot \Big\{ \sum_{j=1}^{N+1} \mathbf{q}_j(\mathbf{x},\mathbf{W})(\mathbf{y}_j - \mathbf{y}_{i^*}) \Big\} \Bigg], \tag{C.4}
$$

$$
\nabla_{\mathbf{W}_{13}} L(\mathbf{W}) = \mathbb{E}\Bigg[ \Big\{ \sum_{j=1}^{N+1} \mathbf{q}_j(\mathbf{x},\mathbf{W})(\mathbf{y}_j)\mathbf{x}_j - \big\{ \sum_{j=1}^{N+1} \mathbf{q}_j(\mathbf{x},\mathbf{W})(\mathbf{x}_j) \big\} \big\{ \sum_{j=1}^{N+1} \mathbf{q}_j(\mathbf{x},\mathbf{W})(\mathbf{y}_j) \big\} \Big\}
$$
$$
\cdot \Big\{ \sum_{j=1}^{N+1} \mathbf{q}_j(\mathbf{x},\mathbf{W})(\mathbf{y}_j - \mathbf{y}_{i^*}) \Big\} \Bigg], \tag{C.5}
$$

$$
\nabla_{\mathbf{W}_{23}} L(\mathbf{W}) = \mathbb{E}\Bigg[ \Big\{ \sum_{j=1}^{N} \mathbf{q}_j(\mathbf{x},\mathbf{W})(\mathbf{y}_j)\mathbf{y}_j - \big\{ \sum_{j=1}^{N} \mathbf{q}_j(\mathbf{x},\mathbf{W})\mathbf{y}_j \big\} \big\{ \sum_{j=1}^{N+1} \mathbf{q}_j(\mathbf{x},\mathbf{W})(\mathbf{y}_j) \big\} \Big\} \cdot \Big\{ \sum_{j=1}^{N+1} \mathbf{q}_j(\mathbf{x},\mathbf{W})(\mathbf{y}_j - \mathbf{y}_{i^*}) \Big\} \Bigg],
$$
$$
\tag{C.6}
$$

$$
\nabla_{\mathbf{W}_{31}} L(\mathbf{W}) = \mathbb{E}\Bigg[ \Big\{ -\big\{ \sum_{j=1}^{N+1} \mathbf{q}_j(\mathbf{x},\mathbf{W})(\mathbf{y}_j) \big\} \cdot \mathbf{q}_{N+1}(\mathbf{W})\mathbf{x}_{N+1}^\top \Big\} \cdot \Big\{ \sum_{j=1}^{N+1} \mathbf{q}_j(\mathbf{x},\mathbf{W})(\mathbf{y}_j - \mathbf{y}_{i^*}) \Big\} \Bigg],
$$
$$
\tag{C.7}
$$

and

$$
\nabla_{\mathbf{W}_{33}} L(\mathbf{W}) = \mathbb{E}\Bigg[ \Big( -\sum_{j=1}^{N+1} \mathbf{q}_j(\mathbf{x},\mathbf{W})\mathbf{y}_j \Big) \mathbf{q}_{N+1}(\mathbf{W}) \cdot \Big\{ \sum_{j=1}^{N+1} \mathbf{q}_j(\mathbf{x},\mathbf{W})(\mathbf{y}_j - \mathbf{y}_{i^*}) \Big\} \Bigg]. \tag{C.8}
$$

By the induction assumption, we have

$$
\mathbf{q}_j(\mathbf{x},\mathbf{W}^k) = \frac{\mathbb{1}_{j\neq N+1}\exp(\xi_1^k \langle \mathbf{x}_j, \mathbf{x}_{N+1}\rangle) + \mathbb{1}_{j=N+1}\exp(\xi_1^k \|\mathbf{x}_{N+1}\|_2^2 - \xi_2^k)}{\sum_{i=1}^{N}\exp(\xi_1^k \langle \mathbf{x}_i, \mathbf{x}_{N+1}\rangle) + \exp(\xi_1^k \|\mathbf{x}_{N+1}\|_2^2 - \xi_2^k)}
$$
$$
= \frac{\mathbb{1}_{j\neq N+1}\exp(\xi_1^k \langle \mathbf{x}_j, \mathbf{x}_{N+1}\rangle) + \mathbb{1}_{j=N+1}\exp(\xi_1^k - \xi_2^k)}{\sum_{i=1}^{N}\exp(\xi_1^k \langle \mathbf{x}_i, \mathbf{x}_{N+1}\rangle) + \exp(\xi_1^k - \xi_2^k)} \tag{C.9}
$$

where the first equality comes from $\|\mathbf{x}_{N+1}\|_2 = 1$. Therefore, under our induction $\mathbf{q}_j$ is only a function of $\{\mathbf{x}_i\}_{i\in[N+1]}$ and $\mathbf{W}^k$, independent to $\{\mathbf{y}_i\}_{i\in[N]}$. Now, with the closed form of $\nabla_{\mathbf{W}_{ij}} L(\mathbf{W})$, we can make the following calculation. First, we calculate $\nabla_{\mathbf{W}_{11}} L(\mathbf{W}^k)$:

$$
\nabla_{\mathbf{W}_{11}} L(\mathbf{W}^k) = \mathbb{E}\Bigg[ \Big\{ \sum_{j=1}^{N+1} \mathbf{q}_j(\mathbf{x},\mathbf{W}^k)(\mathbf{x}_j \mathbf{x}_{N+1}^\top)(\mathbf{y}_j - \mathbf{y}_{i^*}) - \big\{ \sum_{j=1}^{N+1} \mathbf{q}_j(\mathbf{x},\mathbf{W}^k)(\mathbf{x}_j \mathbf{x}_{N+1}^\top) \big\} \big\{ \sum_{j=1}^{N+1} \mathbf{q}_j(\mathbf{x},\mathbf{W}^k)(\mathbf{y}_j - \mathbf{y}_{i^*}) \big\} \Big\}
$$
$$
\cdot \Big\{ \sum_{j=1}^{N+1} \mathbf{q}_j(\mathbf{x},\mathbf{W}^k)(\mathbf{y}_j - \mathbf{y}_{i^*}) \Big\} \Bigg]
$$

$$= \mathbb{E}\Big[\underbrace{\Big\{ \sum_{j=1}^{N+1} \mathbf{q}_j(\mathbf{x}, \mathbf{W}^k)(\mathbf{x}_j\mathbf{x}_{N+1}^\top)(\mathbf{y}_j - \mathbf{y}_i^*)\Big\}\Big\{ \sum_{j=1}^{N+1} \mathbf{q}_j(\mathbf{x}, \mathbf{W}^k)(\mathbf{y}_j - \mathbf{y}_i^*)\Big\}}_{(1)}\Big]$$

$$- \mathbb{E}\Big[\underbrace{\Big\{ \sum_{j=1}^{N+1} \mathbf{q}_j(\mathbf{x}, \mathbf{W}^k)\mathbf{x}_j\mathbf{x}_{N+1}^\top\Big\}\Big\{ \sum_{j=1}^{N+1} \mathbf{q}_j(\mathbf{x}, \mathbf{W}^k)(\mathbf{y}_j - \mathbf{y}_i^*)\Big\}^2}_{(2)}\Big],$$

since $\mathbf{q}(\mathbf{x}, \mathbf{W}^k)$ is only a function of $\mathbf{x}$ by our induction assumption, we have

$$(1) = \mathbb{E}\Big[\underbrace{\Big\{ \sum_{j=1}^{N+1} \mathbf{q}_j(\mathbf{x}, \mathbf{W}^k)(\mathbf{x}_j\mathbf{x}_{N+1}^\top)\mathbf{y}_j\Big\}\Big\{ \sum_{j=1}^{N+1} \mathbf{q}_j(\mathbf{x}, \mathbf{W}^k)\mathbf{y}_j\Big\}}_{(i)}\Big]$$

$$- \mathbb{E}\Big[\underbrace{\Big\{ \sum_{j=1}^{N+1} \mathbf{q}_j(\mathbf{x}, \mathbf{W}^k)(\mathbf{x}_j\mathbf{x}_{N+1}^\top)\mathbf{y}_j\Big\}\mathbf{y}_{i^*}}_{(ii)}\Big] - \mathbb{E}\Big[\underbrace{\Big\{ \sum_{j=1}^{N+1} \mathbf{q}_j(\mathbf{x}, \mathbf{W}^k)(\mathbf{x}_j\mathbf{x}_{N+1}^\top)\mathbf{y}_{i^*}\Big\}\Big\{ \sum_{j=1}^{N+1} \mathbf{q}_j(\mathbf{x}, \mathbf{W}^k)\mathbf{y}_j\Big\}}_{(iii)}\Big]$$

$$+ \mathbb{E}\Big[\underbrace{\sum_{j=1}^{N+1} \mathbf{q}_j(\mathbf{x}, \mathbf{W}^k)(\mathbf{x}_j\mathbf{x}_{N+1}^\top)(\mathbf{y}_{i^*})^2}_{(iv)}\Big],$$

here

$$(i) = \mathbb{E}\Big[\sum_{j=1}^{N+1} \mathbf{q}_j(\mathbf{x}, \mathbf{W}^k)(\mathbf{x}_j\mathbf{x}_{N+1}^\top)\Big(\mathbf{y}_j \sum_{j'=1}^{N+1} \mathbf{q}_j(\mathbf{x}, \mathbf{W}^k)\mathbf{y}_{j'}\Big)\Big]$$

$$= \mathbb{E}\Big[\sum_{j=1}^{N} \mathbf{q}_j(\mathbf{x}, \mathbf{W}^k)^2(\mathbf{x}_j\mathbf{x}_{N+1}^\top)\Big],$$

$$(ii) = -\mathbb{E}\Big[\sum_{j=1}^{N+1} \mathbf{q}_j(\mathbf{x}, \mathbf{W}^k)(\mathbf{x}_j\mathbf{x}_{N+1}^\top)\mathbb{E}\Big[\mathbf{y}_j\mathbf{y}_{i^*}\Big|\mathbf{x}_{1:N+1}\Big]\Big]$$

$$= -\mathbb{E}\Big[\sum_{j=1}^{N} \mathbf{q}_j(\mathbf{x}, \mathbf{W}^k)(\mathbf{x}_j\mathbf{x}_{N+1}^\top) \sum_{j'=1}^{N+1} \mathbb{1}_{j'=\mathrm{argmax}_{i\in[N]}\langle x_{N+1}, x_i\rangle} \mathbb{E}[\mathbf{y}_{j'}\mathbf{y}_j|\mathbf{x}_{1:N+1}]\Big]$$

$$= -\mathbb{E}\Big[\mathbf{q}_{i^*}(\mathbf{x}, \mathbf{W}^k)\mathbf{x}_{i^*}\mathbf{x}_{N+1}^\top\Big],$$

$$(iii) = -\mathbb{E}\Big[\sum_{j,j'=1}^{N+1} \mathbf{q}_j(\mathbf{x}, \mathbf{W}^k)\mathbf{q}_{j'}(\mathbf{x}, \mathbf{W}^k)(\mathbf{x}_j\mathbf{x}_{N+1}^\top)\mathbf{y}_{i^*}\mathbf{y}_{j'}\Big]$$

$$= -\mathbb{E}\Big[\sum_{j=1}^{N+1} \mathbf{q}_j(\mathbf{x}, \mathbf{W}^k)\mathbf{q}_{i^*}(\mathbf{x}, \mathbf{W}^k)(\mathbf{x}_j\mathbf{x}_{N+1}^\top)\Big]$$

$$(iv) = \mathbb{E}\Big[\sum_{j=1}^{N+1} \mathbf{q}_j(\mathbf{x}, \mathbf{W}^k)(\mathbf{x}_j\mathbf{x}_{N+1}^\top)\Big],$$

also for (2), we have

$$(2) = -\mathbb{E}\left[\left\{\sum_{j=1}^{N+1}\mathbf{q}_j(\mathbf{x},\mathbf{W}^k)\mathbf{x}_j\mathbf{x}_{N+1}^\top\right\}\mathbb{E}\left[\left\{\sum_{j=1}^{N+1}\mathbf{q}_j(\mathbf{x},\mathbf{W}^k)(\mathbf{y}_j-\mathbf{y}_{i^*})\right\}^2\Big|\mathbf{x}_{1:N+1}\right]\right]$$

$$= -\mathbb{E}\left[\left\{\sum_{j=1}^{N+1}\mathbf{q}_j(\mathbf{x},\mathbf{W}^k)\mathbf{x}_j\mathbf{x}_{N+1}^\top\right\}\left\{\sum_{j,j'=1}^{N+1}\mathbf{q}_j(\mathbf{x},\mathbf{W}^k)\mathbf{q}_{j'}(\mathbf{x},\mathbf{W}^k)\mathbb{E}\left[\{\mathbf{y}_j\mathbf{y}_{j'}-(\mathbf{y}_j+\mathbf{y}_{j'})\mathbf{y}_{i^*}+\mathbf{y}_{i^*}^2\}\Big|\mathbf{x}_{1:N+1}\right]\right\}\right]$$

$$= -\mathbb{E}\left[\left\{\sum_{j=1}^{N+1}\mathbf{q}_j(\mathbf{x},\mathbf{W}^k)\mathbf{x}_j\mathbf{x}_{N+1}^\top\right\}\left\{1+\sum_{j=1}^{N}\mathbf{q}_j^2(\mathbf{x},\mathbf{W}^k)-\sum_{j,j'=1}^{N+1}\mathbf{q}_j(\mathbf{x},\mathbf{W}^k)\mathbf{q}_{j'}(\mathbf{x},\mathbf{W}^k)\mathbb{E}\left[(\mathbf{y}_j+\mathbf{y}_{j'})\mathbf{y}_{i^*}\Big|\mathbf{x}_{1:N+1}\right]\right\}\right]$$

$$= -\mathbb{E}\left[\left\{\sum_{j=1}^{N+1}\mathbf{q}_j(\mathbf{x},\mathbf{W}^k)\mathbf{x}_j\mathbf{x}_{N+1}^\top\right\}\left\{1+\sum_{j=1}^{N}\mathbf{q}_j^2(\mathbf{x},\mathbf{W}^k)-2\mathbf{q}_{i^*}(\mathbf{x},\mathbf{W}^k)\right\}\right]$$

(1)+(2) gives us

$$\nabla_{\mathbf{W}_{11}}L(\mathbf{W}^k) = \mathbb{E}\left[\sum_{j=1}^{N}\mathbf{q}_j^2(\mathbf{x},\mathbf{W}^k)(\mathbf{x}_j\mathbf{x}_{N+1}^\top)\right] - \mathbb{E}\left[\mathbf{q}_{i^*}(\mathbf{x},\mathbf{W}^k)(\mathbf{x}_{i^*}\mathbf{x}_{N+1}^\top)\right] + \mathbb{E}\left[\mathbf{q}_{i^*}(\mathbf{x},\mathbf{W}^k)\left(\sum_{j=1}^{N+1}\mathbf{q}_j(\mathbf{x},\mathbf{W}^k)(\mathbf{x}_j\mathbf{x}_{N+1}^\top)\right)\right.$$

$$- \mathbb{E}\left[\sum_{j=1}^{N}\mathbf{q}_j^2(\mathbf{x},\mathbf{W}^k)\left(\sum_{j=1}^{N+1}\mathbf{q}_j(\mathbf{x},\mathbf{W}^k)(\mathbf{x}_j\mathbf{x}_{N+1}^\top)\right)\right]. \tag{C.10}$$

For $\nabla_{\mathbf{W}_{21}}L(\mathbf{W}_k)$, we have

$$\nabla_{\mathbf{W}_{21}}L(\mathbf{W}_k) = \mathbb{E}\left[\left\{\sum_{j=1}^{N}\mathbf{q}_j(\mathbf{x},\mathbf{W}_k)(\mathbf{y}_j\mathbf{x}_{N+1}^\top)(\mathbf{y}_j)-\left\{\sum_{j=1}^{N}\mathbf{q}_j(\mathbf{x},\mathbf{W}_k)(\mathbf{y}_j\mathbf{x}_{N+1}^\top)\right\}\left\{\sum_{j=1}^{N+1}\mathbf{q}_j(\mathbf{x},\mathbf{W}_k)\mathbf{y}_j\right\}\right.$$

$$\left.\cdot\left\{\sum_{j=1}^{N+1}\mathbf{q}_j(\mathbf{x},\mathbf{W}_k)(\mathbf{y}_j-\mathbf{y}_{i^*})\right\}\right],$$

note that it is the expectation of multiplication of three linear functions of $\mathbf{y}_i$, therefore $\nabla_{\mathbf{W}_{21}}L(\mathbf{W}_k)$ equals to 0 by symmetry. For $\nabla_{\mathbf{W}_{31}}L(\mathbf{W}_k)$, we have

$$\nabla_{\mathbf{W}_{31}}L(\mathbf{W}_k) = \mathbb{E}\left[\left\{-\left\{\sum_{j=1}^{N+1}\mathbf{q}_j(\mathbf{x},\mathbf{W}_k)(\mathbf{y}_j)\right\}\cdot\mathbf{q}_{N+1}(\mathbf{x},\mathbf{W}_k)\mathbf{x}_{N+1}^\top\right\}\cdot\left\{\sum_{j=1}^{N+1}\mathbf{q}_j(\mathbf{W}_k)(\mathbf{y}_j-\mathbf{y}_{i^*})\right\}\right]$$

$$= 0,$$

here the second equality comes from $\mathbf{q}_j(\mathbf{x},\mathbf{W}_k)$ being an even function of $\mathbf{x}$ and $\mathbf{x}$ has a symmetric distribution. For $\nabla_{\mathbf{W}_{13}}L(\mathbf{W}_k)$, we have

$$\nabla_{\mathbf{W}_{13}}L(\mathbf{W}_k) = \mathbb{E}\left[\left\{\sum_{j=1}^{N+1}\mathbf{q}_j(\mathbf{W}^{KQ})(\mathbf{y}_j)\mathbf{x}_j-\left\{\sum_{j=1}^{N+1}\mathbf{q}_j(\mathbf{W}^{KQ})(\mathbf{x}_j)\right\}\left\{\sum_{j=1}^{N+1}\mathbf{q}_j(\mathbf{W}^{KQ})(\mathbf{y}_j)\right\}\right.$$

$$\left.\cdot\left\{\sum_{j=1}^{N+1}\mathbf{q}_j(\mathbf{W}^{KQ})(\mathbf{y}_j-\mathbf{y}_{i^*})\right\}\right]$$

$$= 0$$

here the second equality comes from $\mathbf{q}_j(x,\mathbf{W}_k)$ is a function of $\|\mathbf{x}\|^2$ and $\mathbf{x}$ has a symmetric distribution. For $\nabla_{\mathbf{W}_{23}}L(\mathbf{W}_k)$, we have

$$\nabla_{\mathbf{W}_{23}}L(\mathbf{W}_k) = \left\{\sum_{j=1}^{N}\mathbf{q}_j(\mathbf{W}^{KQ})(\mathbf{y}_j)\mathbf{y}_j-\left\{\sum_{j=1}^{N}\mathbf{q}_j(\mathbf{W}^{KQ})\mathbf{y}_j\right\}\left\{\sum_{j=1}^{N+1}\mathbf{q}_j(\mathbf{W}^{KQ})(\mathbf{y}_j)\right\}\right\}$$

$$
\cdot \left\{ \sum_{j=1}^{N+1} \mathbf{q}_j(\mathbf{W}^{KQ})(\mathbf{y}_j - \mathbf{y}_{bi^*}) \right\}
$$

$$
= 0,
$$

due to its symmetry in $\mathbf{y}$. For $\nabla_{\mathbf{W}_{33}} L(\mathbf{W}_k)$, we have the following calculation:

$$
\nabla_{\mathbf{W}_{33}} L(\mathbf{W}_k) = \mathbb{E}\left[ \left( -\sum_{j=1}^{N+1} \mathbf{q}_j(\mathbf{x}, \mathbf{W}_k)\mathbf{y}_j \right) \mathbf{q}_{N+1}(\mathbf{x}, \mathbf{W}_k) \cdot \left\{ \sum_{j=1}^{N+1} \mathbf{q}_j(\mathbf{x}, \mathbf{W}_k)(\mathbf{y}_j - \mathbf{y}_{i^*}) \right\} \right]
$$

$$
= \mathbb{E}\left[ \mathbf{q}_{N+1}(\mathbf{x}, \mathbf{W}_k) \left\{ \mathbf{y}_{i^*} \left( \sum_{j=1}^{N+1} \mathbf{q}_j(\mathbf{x}, \mathbf{W}_k)\mathbf{y}_j \right) - \left( \sum_{j=1}^{N+1} \mathbf{q}_j(\mathbf{x}, \mathbf{W}_k)\mathbf{y}_j \right)^2 \right\} \right]
$$

$$
= \mathbb{E}\left[ \mathbf{q}_{N+1}(\mathbf{x}, \mathbf{W}_k) \left\{ \mathbf{q}_{i^*}(\mathbf{x}, \mathbf{W}_k) - \sum_{j=1}^{N} \mathbf{q}_j^2(\mathbf{x}, \mathbf{W}_k) \right\} \right] \tag{C.11}
$$

With all previous calculations, we know that $\nabla_{\mathbf{W}_{ij}} L(\mathbf{W}^k)$ equals 0 except for $\nabla_{\mathbf{W}_{11}} L(\mathbf{W}^k)$ and $\nabla_{\mathbf{W}_{33}} L(\mathbf{W}^k)$ with our induction assumption.

Now we only need to prove that $\nabla_{\mathbf{W}_{11}} L(\mathbf{W})$ is a diagonal matrix. Recall that $\{\mathbf{x}_i\}_{i \in [N+1]}$ is identically uniformly distributed on $\mathbb{S}^{d-1}$, thus it naturally satisfies the following Assumption:

**Assumption 4** (Rotational Invariance). *We say the distribution of $\mathbf{x}$ is rotationally invariance, if $\mathbf{x} \sim \mathrm{P}_x$, and for every $U \in \mathcal{O}(d)$, where $\mathcal{O}$ is the group of orthogonal matrices, we have $\mathrm{P}_x = U_{\#}\mathrm{P}_x$.*

A straightforward lemma under Assumption 4 is the following.

**Lemma 6.** *Suppose $\{\mathbf{x}_i\}_{i \in [N+1]}$ are sampled i.i.d. from $\mathrm{P}_x$ and Assumption 4 holds, then we have $\bigotimes_{i=1}^{N+1} \mathrm{P}_{\mathbf{x}_i} = \bigotimes_{i=1}^{N+1} \mathrm{P}_{U\mathbf{x}_i}$ for all $U \in \mathcal{O}(d)$.*

*Proof.* Since $\mathbf{x}_{i^*}$ are sampled i.i.d. from $\mathrm{P}_x$, we have

$$
\bigotimes_{i=1}^{N+1} \mathrm{P}_{\mathbf{x}_i} = \bigotimes_{i=1}^{N+1} \mathrm{P}_{U\mathbf{x}_i}.
$$

$\square$

**Lemma 7.** *When $\mathbf{x}_i \sim \mathrm{P}_x$, $\mathrm{P}_x$ satisfies Assumption 4, and $\mathbf{W}^k$ is a diagonal matrix with $[\mathbf{W}^k]_{[d] \times [d]} = c_k I_d$, we have $\mathbb{E}[\nabla_{\mathbf{W}_{11}} L(\mathbf{W}^k)] = a_k \cdot I_d$.*

*Proof.* Note that by the induction assumption, we have

$$
\mathbf{q}_j(\mathbf{x}, \mathbf{W}^k) = \frac{\mathbb{1}_{j \neq N+1} \exp(\xi_1^k \langle \mathbf{x}_j, \mathbf{x}_{N+1}\rangle) + \mathbb{1}_{j=N+1} \exp(\xi_1^k - \xi_2^k)}{\sum_{i=1}^{N} \exp(\xi_1^k \langle \mathbf{x}_i, \mathbf{x}_{N+1}\rangle) + \exp(\xi_1^k - \xi_2^k)},
$$

therefore, $\mathbf{q}_j(\mathbf{x}, \mathbf{W}^k) = \mathbf{q}_j(U\mathbf{x}, \mathbf{W}^k)$ for all $U \in \mathcal{O}(d)$. By Eq. (C.10), there exits a set of functions

$$
\{g_i^k(x)\}_{i \in [N]} \cup \{g_{i^*}^k(x)\} : \mathbb{R} \to \mathbb{R}
$$

such that

$$
\nabla_{\mathbf{W}_{11}^k} L(\mathbf{W}) = \sum_{i=1}^{N} g_i^k(\mathbf{x}_i^\top \mathbf{x}_{N+1}) \cdot \mathbf{x}_i \mathbf{x}_{N+1}^\top + g_{i^*}^k(\mathbf{x}_{i^*}^\top \mathbf{x}_{N+1}) \cdot \mathbf{x}_{i^*} \mathbf{x}_{N+1}^\top,
$$

here the expectation is taken with respect to $\{\mathbf{x}_i\}_{i \in [N+1]}$. By Lemma 6, we have $\mathrm{P}_{\mathbf{x}_i} \otimes \mathrm{P}_{\mathbf{x}_j} = \mathrm{P}_{U\mathbf{x}_i} \otimes \mathrm{P}_{U\mathbf{x}_j}$ for any orthogonal matrix $U \in \mathcal{O}(d)$, therefore

$$
\mathbb{E}[g_i^k(\mathbf{x}_i^\top \mathbf{x}_{N+1}) \cdot \mathbf{x}_i \mathbf{x}_{N+1}] = \mathbb{E}[g_i^k((U\mathbf{x}_i)^\top (U\mathbf{x}_{N+1})) \cdot (U\mathbf{x}_i)(U\mathbf{x}_{N+1})^\top]
$$

$$
= U\mathbb{E}[g_i^k(\mathbf{x}_i^\top \mathbf{x}_{N+1}) \cdot \mathbf{x}_i \mathbf{x}_{N+1}^\top]U^\top,
$$

here the first inequality comes from $P_{\mathbf{x}_i} \otimes P_{\mathbf{x}_j} = P_{U\mathbf{x}_i} \otimes P_{U\mathbf{x}_j}$. Such identity holds for all orthogonal matrix $U$, therefore $\mathbb{E}[g_i^k(\mathbf{x}_i^\top \mathbf{x}_{N+1}) \cdot \mathbf{x}_i \mathbf{x}_{N+1}]$ must be a multiplication of the identity matrix $I_d$. Now, note that for all orthogonal matrix $U$,

$$\mathbf{x}_{i^*} = \underset{\mathbf{x}_i \in \{\mathbf{x}_j\}_{j \in [N]}}{\operatorname{argmin}} \|\mathbf{x}_i - \mathbf{x}_{N+1}\|_2,$$

$$U\mathbf{x}_{i^*} = U \underset{\mathbf{x}_i \in \{\mathbf{x}_j\}_{j \in [N]}}{\operatorname{argmin}} \|U\mathbf{x}_i - U\mathbf{x}_{N+1}\|_2,$$

therefore

$$P(\mathbf{x}_{i^*}^\top \mathbf{x}_{N+1} | \mathbf{x}_{1:(N+1)}) = P(U\mathbf{x}_{i^*} \mathbf{x}_{N+1}^\top U^\top | U\mathbf{x}_{1:N+1}).$$

Since $P(\mathbf{x}_{1:N+1}) = P(U\mathbf{x}_{1:N+1})$, we have $P(\mathbf{x}_{i^*}^\top \mathbf{x}_{N+1}) = P(U\mathbf{x}_{i^*}\mathbf{x}_{N+1}^\top U^\top)$, therefore

$$\mathbb{E}[g_{i^*}^k(\mathbf{x}_{i^*}^\top \mathbf{x}_{N+1}) \cdot \mathbf{x}_{i^*}\mathbf{x}_{N+1}^\top] = \mathbb{E}[g_{i^*}^k((U\mathbf{x}_{i^*})^\top(U\mathbf{x}_{N+1})) \cdot (U\mathbf{x}_{i^*})(U\mathbf{x}_{N+1})^\top]$$
$$= U\mathbb{E}[g_{i^*}^k(\mathbf{x}_{i^*}^\top \mathbf{x}_{N+1}) \cdot \mathbf{x}_{i^*}\mathbf{x}_{N+1}^\top]U^\top.$$

Again, since $U$ is an arbitrary orthogonal matrix, we conclude that $\mathbb{E}[g_{i^*}^k(\mathbf{x}_{i^*}^\top \mathbf{x}_{N+1}) \cdot \mathbf{x}_{i^*}\mathbf{x}_{N+1}^\top]$ is a multiplication of the identity matrix. Summing everything together, we conclude the proof that $\nabla_{\mathbf{W}_{11}^k} L(\mathbf{W}^k)$ is a multiplication of an identity matrix. $\qquad\square$

With Lemma 7, Eq. (2.6), and the previous calculations, we conclude our induction.

## C.2 Evolution of $\xi_1$ and $\xi_2$

In this section, we characterize the training dynamics of $\xi_1$ and $\xi_2$ under gradient descent. Our results can be broken down into the following steps:

(1) We prove that when $\xi_1^0$ and $\xi_2^0$ are initialized as in Assumption 2 , with $\sigma$, $N$, $d$ satisfy the condition in Theorem 1, we have $\xi_1^1 - \xi_1^0 \geq 0$.

(2) We provide an upper bound and lower bound for $\xi_1^{k+1} - \xi_1^k$ and $\xi_2^{k+1} - \xi_2^k$.

(3) We prove that there exits constant $c_1 \in (0,1)$ such that $\xi_1^k \in (0, c_1\xi_2^k)$.

(4) We prove that $\xi_2^{k+1} - \xi_2^k = \Omega(\eta \cdot \exp(-\text{poly}(N,d) \cdot \xi_2^k))$, we thus conclude that $\xi_2^k = \Omega(\eta \cdot \log(\text{poly}(N,d)) \cdot \log(k))$.

Combining (3) and (4), we conclude this section by proving that $\xi_1^k$, $\xi_2^k$ and $\xi_2^k - \xi_1^k$ are both of scale $\Omega(\log k)$.

**Step (1): Initial incrementation of $\xi_1^0$.** First, we prove that $\xi_1^1 - \xi_1^0 \geq 0$. Note that by Lemma 7 and Eq. (C.10), we have

$$\frac{d}{\eta}(\xi_1^0 - \xi_1^1) = \nabla_{\mathbf{W}_{11}} L(\mathbf{W}^0)$$

$$= \mathbb{E}\left[\sum_{j=1}^N \mathbf{q}_j^2(\mathbf{x}, \mathbf{W}^0)(\mathbf{x}_j^\top \mathbf{x}_{N+1})\right] - \mathbb{E}\left[\mathbf{q}_{i^*}(\mathbf{x}, \mathbf{W}^0)(\mathbf{x}_{i^*}^\top \mathbf{x}_{N+1})\right]$$

$$+ \mathbb{E}\left[\mathbf{q}_{i^*}(\mathbf{x}, \mathbf{W}^0)\left(\sum_{j=1}^{N+1} \mathbf{q}_j(\mathbf{x}, \mathbf{W}^0)(\mathbf{x}_j^\top \mathbf{x}_{N+1})\right)\right]$$

$$- \mathbb{E}\left[\sum_{j=1}^N \mathbf{q}_j^2(\mathbf{x}, \mathbf{W}^0)\left(\sum_{j=1}^{N+1} \mathbf{q}_j(\mathbf{x}, \mathbf{W}^0)(\mathbf{x}_j^\top \mathbf{x}_{N+1})\right)\right]$$

$$\leq -\frac{1}{N+1}\mathbb{E}[\mathbf{x}_{i^*}\mathbf{x}_{N+1}^\top] + \frac{1}{(N+1)^3}\mathbb{E}[\mathbf{x}\mathbf{x}^\top], \qquad (\xi_2^0 \geq 0)$$

To prove that $\xi_1^0 - \xi_1^1 < 0$, we only need

$$-\frac{1}{N+1}\mathbb{E}[\mathbf{x}_{i^*}\mathbf{x}_{N+1}^\top] + \frac{1}{(N+1)^3}\mathbb{E}[\mathbf{x}\mathbf{x}^\top] \leq 0.$$

However, by Lemma 18, this can be guaranteed by $N \geq O(\sqrt{d}\log d)$.

**Step (2): Scale of $\xi_1^{k+1} - \xi_1^k$ and $\xi_2^{k+1} - \xi_2^k$**  Note that in every gradient update with stepsize $\eta$, we have

$$\frac{d}{\eta}(\xi_1^{k+1} - \xi_1^k) = \mathbb{E}\left[\mathbf{q}_{i^*}(\mathbf{x}, \mathbf{W})(\mathbf{x}_{i^*}^\top\mathbf{x}_{N+1})\right] - \mathbb{E}\left[\sum_{j=1}^N \mathbf{q}_j^2(\mathbf{x}, \mathbf{W})(\mathbf{x}_j^\top\mathbf{x}_{N+1})\right]$$

$$- \mathbb{E}\left[\left\{\mathbf{q}_{i^*}(\mathbf{x}, \mathbf{W}) - \sum_{j=1}^N \mathbf{q}_j^2(\mathbf{x}, \mathbf{W})\right\}\left(\sum_{j=1}^{N+1} \mathbf{q}_j(\mathbf{x}, \mathbf{W})(\mathbf{x}_j^\top\mathbf{x}_{N+1})\right)\right]$$

$$\tag{C.12}$$

$$= \mathbb{E}\left[\left\{\mathbf{q}_{i^*}(\mathbf{x}, \mathbf{W}) - \sum_{j=1}^N \mathbf{q}_j^2(\mathbf{x}, \mathbf{W})\right\}\left\{\mathbf{x}_{i^*}^\top\mathbf{x}_{N+1} - \sum_{j=1}^{N+1} \mathbf{q}_j(\mathbf{x}, \mathbf{W})(\mathbf{x}_j^\top\mathbf{x}_{N+1})\right\}\right]$$

$$+ \mathbb{E}\left[\sum_{j=1}^N \mathbf{q}_j^2(\mathbf{x}, \mathbf{W})\left\{\mathbf{x}_{i^*}^\top\mathbf{x}_{N+1} - \mathbf{x}_j^\top\mathbf{x}_{N+1}\right\}\right] \tag{C.13}$$

we also have the following estimation for the update of $\xi_3$:

$$\frac{1}{\eta}(\xi_2^{k+1} - \xi_2^k) = \mathbb{E}[\mathbf{q}_{N+1}(\mathbf{x}, \mathbf{W}^k)\{\mathbf{q}_{i^*}(\mathbf{x}, \mathbf{W}^k) - \sum_{j=1}^N \mathbf{q}_j^2(\mathbf{x}, \mathbf{W}^k)\}]$$

$$\geq \mathbb{E}\left[\mathbf{q}_{N+1}(\mathbf{x}, \mathbf{W}^k) \cdot \left\{\mathbf{q}_{i^*}(\mathbf{x}, \mathbf{W}^k) - \mathbf{q}_{i^*}(\mathbf{x}, \mathbf{W}^k) \cdot \sum_{j=1}^N \mathbf{q}_j(\mathbf{x}, \mathbf{W}^k)\right\}\right]$$

$$= \mathbb{E}[\mathbf{q}_{i^*}(\mathbf{x}, \mathbf{W}^k) \cdot \mathbf{q}_{N+1}^2(\mathbf{x}, \mathbf{W}^k)], \tag{C.14}$$

which shows that in each iteration, $\xi_3$ will at least increment by a scale of $\eta \cdot \mathbb{E}[\mathbf{q}_{i^*}(\mathbf{x}, \mathbf{W}^k) \cdot \mathbf{q}_{N+1}^2(\mathbf{x}, \mathbf{W}^k)]$. Combining Eq. (C.12), (C.11), we have the following estimation:

**Lemma 8.** *We have*

$$\frac{1}{\eta} \cdot \left\{d(\xi_1^{k+1} - \xi_1^k) + 2(\xi_2^{k+1} - \xi_2^k)\right\} \geq \left(1 - \frac{1}{2^N}\right)C_d\exp(-6\xi_1^k)$$

*for all $k \geq 0$.*

*Proof.* First, note that we have the following relations:

$$\frac{1}{\eta} \cdot \left\{d(\xi_1^{k+1} - \xi_1^k) + 2(\xi_2^{k+1} - \xi_2^k)\right\}$$

$$= \mathbb{E}\left[\mathbf{q}_{i^*}(\mathbf{x}, \mathbf{W})(\mathbf{x}_{i^*}^\top\mathbf{x}_{N+1})\right] - \mathbb{E}\left[\sum_{j=1}^N \mathbf{q}_j^2(\mathbf{x}, \mathbf{W})(\mathbf{x}_j^\top\mathbf{x}_{N+1})\right]$$

$$+ \mathbb{E}[\mathbf{q}_{N+1}(\mathbf{x}, \mathbf{W}^k)\{\mathbf{q}_{i^*}(\mathbf{x}, \mathbf{W}^k) - \sum_{j=1}^N \mathbf{q}_j^2(\mathbf{x}, \mathbf{W}^k)\}]$$

$$- \mathbb{E}\left[\left\{\mathbf{q}_{i^*}(\mathbf{x}, \mathbf{W}) - \sum_{j=1}^N \mathbf{q}_j^2(\mathbf{x}, \mathbf{W})\right\}\left(\sum_{j=1}^N \mathbf{q}_j(\mathbf{x}, \mathbf{W})(\mathbf{x}_j^\top\mathbf{x}_{N+1})\right)\right]$$

$$= \mathbb{E}\left[\{\mathbf{q}_{i^*}(\mathbf{x}, \mathbf{W}) - \sum_{j=1}^{N} \mathbf{q}_j^2(\mathbf{x}, \mathbf{W})\}\{(\mathbf{x}_{i^*}^\top \mathbf{x}_{N+1} + 1) - \sum_{j=1}^{N} \mathbf{q}_j(\mathbf{x}, \mathbf{W})(\mathbf{x}_j^\top \mathbf{x}_{N+1})\}\right]$$

$$+ \mathbb{E}\left[\sum_{j=1}^{N} \mathbf{q}_j^2(\mathbf{x}, \mathbf{W})\{\mathbf{x}_{i^*}^\top \mathbf{x}_{N+1} - \mathbf{x}_j^\top \mathbf{x}_{N+1}\}\right]$$

$$\geq \mathbb{E}\left[\{\mathbf{q}_{i^*}(\mathbf{x}, \mathbf{W}) - \sum_{j=1}^{N} \mathbf{q}_j^2(\mathbf{x}, \mathbf{W})\} \cdot \mathbf{q}_{N+1}(\mathbf{x}, \mathbf{W})(\mathbf{x}_{i^*}^\top \mathbf{x}_{N+1} + 1)\right]$$

$$+ \mathbb{E}\left[\sum_{j=1}^{N} \mathbf{q}_j^2(\mathbf{x}, \mathbf{W})\{\mathbf{x}_{i^*}^\top \mathbf{x}_{N+1} - \mathbf{x}_j^\top \mathbf{x}_{N+1}\}\right], \tag{C.15}$$

here the inequality comes from $\mathbf{x}_{i^*}^\top \mathbf{x}_{N+1} \geq \mathbf{x}_j^\top \mathbf{x}_{N+1}$ for all $j \in [N]$. Moreover, note that when $\xi_1 \geq 0$, we have

$$1 = \sum_{j=1}^{N} \mathbf{q}_j(\mathbf{x}, \mathbf{W}) + \mathbf{q}_{N+1}(\mathbf{x}, \mathbf{W}) \leq N\mathbf{q}_{i^*}(\mathbf{x}, \mathbf{W}) + \mathbf{q}_{N+1}(\mathbf{x}, \mathbf{W}),$$

which is equivalent to $\mathbf{q}_{i^*}(\mathbf{x}, \mathbf{W}) \geq \frac{1 - \mathbf{q}_{N+1}(\mathbf{x}, \mathbf{W})}{N}$. Since

$$\frac{\mathbf{q}_{i^*}(\mathbf{x}, \mathbf{W})}{\mathbf{q}_{N+1}(\mathbf{x}, \mathbf{W})} = \exp\left(\xi_1(\mathbf{x}_{i^*}^\top \mathbf{x}_{N+1} - 1) + \xi_3\right),$$

we have

$$\mathbf{q}_{i^*}(\mathbf{x}, \mathbf{W}) \geq \frac{1}{N + \exp(\xi_1(1 - \mathbf{x}_{i^*}^\top \mathbf{x}_{N+1}) - \xi_3)}. \tag{C.16}$$

Therefore we have

$$\mathbb{E}\left[\sum_{j=1}^{N} \mathbf{q}_j^2(\mathbf{x}, \mathbf{W})\{\mathbf{x}_{i^*}^\top \mathbf{x}_{N+1} - \mathbf{x}_j^\top \mathbf{x}_{N+1}\}\right]$$

$$= \mathbb{E}\left[\mathbf{q}_{i^*}^2(\mathbf{x}, \mathbf{W}^k) \sum_{j=1}^{N} \frac{\mathbf{q}_j^2(\mathbf{x}, \mathbf{W}^k)}{\mathbf{q}_{i^*}^2(\mathbf{x}, \mathbf{W}^k)} \cdot \{\mathbf{x}_{i^*}^\top \mathbf{x}_{N+1} - \mathbf{x}_j^\top \mathbf{x}_{N+1}\}\right]$$

$$\geq \mathbb{E}\left[\left(\frac{1}{N + \exp(\xi_1(1 - \mathbf{x}_{i^*}^\top \mathbf{x}_{N+1}) - \xi_3)}\right)^2 \sum_{j=1}^{N} \exp\left(2\xi_1^k\{\mathbf{x}_j^\top \mathbf{x}_{N+1} - \mathbf{x}_{i^*}^\top \mathbf{x}_{N+1}\}\right)\{\mathbf{x}_{i^*}^\top \mathbf{x}_{N+1} - \mathbf{x}_j^\top \mathbf{x}_{N+1}\}\right]$$

$$\geq \frac{\exp(-4\xi_1^k)}{(N + \exp(\xi_1^k - \xi_2^k))^2} \mathbb{E}\left[\sum_{j=1}^{N} \{\mathbf{x}_{i^*}^\top \mathbf{x}_{N+1} - \mathbf{x}_j^\top \mathbf{x}_{N+1}\}|\mathbf{x}_{i^*}^\top \mathbf{x}_{N+1} \geq 0\right]\mathbb{P}(\mathbf{x}_{i^*}^\top \mathbf{x}_{N+1} \geq 0)$$

$$\geq \left(1 - \frac{1}{2^N}\right)\frac{C_d \exp(-4\xi_1^k)}{(N + \exp(\xi_1^k - \xi_2^k))^2}$$

$$\geq \left(1 - \frac{1}{2^N}\right)C_d \exp(-6\xi_1^k) \tag{C.17}$$

here $C_d$ is a constant that only pertains to $d$. The first inequality comes from Eq. (C.16), and the third inequality comes from Lemma 17. Finally, note that

$$\mathbb{E}\left[\{\mathbf{q}_{i^*}(\mathbf{x}, \mathbf{W}) - \sum_{j=1}^{N} \mathbf{q}_j^2(\mathbf{x}, \mathbf{W})\} \cdot \mathbf{q}_{N+1}(\mathbf{x}, \mathbf{W})(\mathbf{x}_{i^*}^\top \mathbf{x}_{N+1} + 1)\right] \geq 0,$$

and we conclude the proof. $\square$

Next, we provide the following upper bound for the increment of $\xi_1^k$ and $\xi_2^k$ whenever $\xi_1^k \geq 0$.

**Lemma 9.** *When $\xi_1^k \geq 0$ and $N \geq O(\sqrt{d}\log d)$ as in Theorem 1, we have the following inequalities:*

$$\frac{1}{\eta}(\xi_1^{k+1} - \xi_1^k) \leq \frac{2N}{d}\exp\left(-\frac{4}{(N+1)^2}\xi_1^k\right) - \frac{a_{n,d}}{dN^3 e}\exp(2(\xi_1^k - \xi_2^k)),$$

*where $a_{n,d} = \left(2N\sqrt{d}\right)^{-\frac{2}{d-3}}$, and*

$$\frac{1}{\eta}(\xi_2^{k+1} - \xi_2^k) \leq \exp(2\xi_1^k - \xi_2^k).$$

*Proof.* We first prove the first inequality. Recall that

$$\frac{d}{\eta}(\xi_1^{k+1} - \xi_1^k) = \mathbb{E}\left[\left\{\mathbf{q}_{i^*}(\mathbf{x}, \mathbf{W}) - \sum_{j=1}^{N}\mathbf{q}_j^2(\mathbf{x}, \mathbf{W})\right\}\left\{\mathbf{x}_{i^*}^\top\mathbf{x}_{N+1} - \sum_{j=1}^{N+1}\mathbf{q}_j(\mathbf{x}, \mathbf{W})(\mathbf{x}_j^\top\mathbf{x}_{N+1})\right\}\right]$$

$$+ \mathbb{E}\left[\sum_{j=1}^{N}\mathbf{q}_j^2(\mathbf{x}, \mathbf{W})\{\mathbf{x}_{i^*}^\top\mathbf{x}_{N+1} - \mathbf{x}_j^\top\mathbf{x}_{N+1}\}\right],$$

when $\xi_1^k \geq 0$, we have $\mathbf{q}_{i^*}(\mathbf{x}, \mathbf{W}) - \sum_{j=1}^{N}\mathbf{q}_j^2(\mathbf{x}, \mathbf{W}) \geq 0$, therefore

$$\frac{d}{\eta}(\xi_1^{k+1} - \xi_1^k) \leq \underbrace{\mathbb{E}\left[\sum_{j=1}^{N}\mathbf{q}_j^2(\mathbf{x}, \mathbf{W})\{\mathbf{x}_{i^*}^\top\mathbf{x}_{N+1} - \mathbf{x}_j^\top\mathbf{x}_{N+1}\}\right]}_{(i)}$$

$$\underbrace{- \mathbb{E}\left[\left\{\mathbf{q}_{i^*}(\mathbf{x}, \mathbf{W}^k) - \sum_{j=1}^{N}\mathbf{q}_j^2(\mathbf{x}, \mathbf{W}^k)\right\}\mathbf{q}_{N+1}(\mathbf{x}, \mathbf{W}^k)(1 - \mathbf{x}_{i^*}^\top\mathbf{x}_{N+1})\right]}_{(iii)}$$

$$\underbrace{+ \mathbb{E}\left[\left\{\mathbf{q}_{i^*}(\mathbf{x}, \mathbf{W}) - \sum_{j=1}^{N}\mathbf{q}_j^2(\mathbf{x}, \mathbf{W})\right\}\left\{\sum_{j=1}^{N}\mathbf{q}_j(\mathbf{x}, \mathbf{W})(\mathbf{x}_{i^*}^\top\mathbf{x}_{N+1} - \mathbf{x}_j^\top\mathbf{x}_{N+1})\right\}\right]}_{(ii)}$$

We have the following bound for (iii) when $\xi_1^k \geq 0$:

$$-(iii) \geq \frac{1}{e}\frac{1}{\left(2N\sqrt{d}\right)^{\frac{2}{d-3}}} \cdot \mathbb{E}\left[\left\{\mathbf{q}_{i^*}(\mathbf{x}, \mathbf{W}^k) - \sum_{j=1}^{N}\mathbf{q}_j^2(\mathbf{x}, \mathbf{W}^k)\right\}\mathbf{q}_{N+1}(\mathbf{x}, \mathbf{W}^k)\,\Big|\,\mathbf{x}_{i^*}^\top\mathbf{x}_{N+1} \geq 1 - a_{n,d}\right]$$

$$\geq \frac{1}{e}\frac{1}{\left(2N\sqrt{d}\right)^{\frac{2}{d-3}}} \cdot \mathbb{E}\left[\mathbf{q}_{i^*}(\mathbf{x}, \mathbf{W}^k)\mathbf{q}_{N+1}^2(\mathbf{x}, \mathbf{W}^k)\right]$$

$$\geq \frac{1}{e}\frac{1}{\left(2N\sqrt{d}\right)^{\frac{2}{d-3}}} \cdot \frac{1}{N} \cdot \mathbb{E}\left[(1 - \mathbf{q}_{N+1}(\mathbf{x}, \mathbf{W}^k))\mathbf{q}_{N+1}^2(\mathbf{x}, \mathbf{W}^k)\right]$$

$$\geq \frac{1}{N^3}\frac{1}{e}\frac{1}{\left(2N\sqrt{d}\right)^{\frac{2}{d-3}}}\exp(2(\xi_1^k - \xi_2^k))$$

where the first inequality comes from Lemma 19. Now we aim to bound (i) and (ii) separately. First, we have

$$(i) = \mathbb{E}\left[\mathbf{q}_{i^*}^2(\mathbf{x}, \mathbf{W}^k)\sum_{j=1}^{N}\frac{\mathbf{q}_j^2(\mathbf{x}, \mathbf{W}^k)}{\mathbf{q}_{i^*}^2(\mathbf{x}, \mathbf{W}^k)}\{\mathbf{x}_{i^*}^\top\mathbf{x}_{N+1} - \mathbf{x}_j^\top\mathbf{x}_{N+1}\}\right]$$

$$= \mathbb{E}\left[\sum_{j=1}^{N}\exp\left(-2\xi_1^k\{\mathbf{x}_{i^*}^\top\mathbf{x}_{N+1} - \mathbf{x}_j^\top\mathbf{x}_{N+1}\}\right)\{\mathbf{x}_{i^*}^\top\mathbf{x}_{N+1} - \mathbf{x}_j^\top\mathbf{x}_{N+1}\}\right]$$

$$\leq N \exp\left(-2\xi_1^k \mathbb{E}[\mathbf{x}_{i^*}^\top \mathbf{x}_{N+1} - \mathbf{x}_j^\top \mathbf{x}_{N+1}]\right) \mathbb{E}[\mathbf{x}_{i^*}^\top \mathbf{x}_{N+1} - \mathbf{x}_j^\top \mathbf{x}_{N+1}]$$
$$= N \exp(-2\xi_1^k \mathbb{E}[\mathbf{x}_{i^*}^\top \mathbf{x}_{N+1}]) \mathbb{E}[\mathbf{x}_{i^*}^\top \mathbf{x}_{N+1}]$$
$$\leq N \exp\left(-\frac{4}{(N+1)^2}\xi_1^k\right) \tag{C.18}$$

here the first inequality comes from Jensen's inequality and the convexity of $x\exp(-x)$ between $[0,2]$, and the second one comes from $\mathbb{E}[\mathbf{x}_{i^*}^\top \mathbf{x}_{N+1}] \geq \frac{2}{(N+1)^2}$, which is guaranteed by the condition of $N$ in Theorem 1 and Lemma 18. Thus we conclude the proof for the first inequality. For (ii), denote the second largest order statistics in $\{\mathbf{x}_i^\top \mathbf{x}_{N+1}\}_{i\in[N]}$ by $\mathbf{x}_{(2)}^\top \mathbf{x}_{N+1}$, we have the following inequalities:

$$\text{(ii)} \leq \mathbb{E}\left[\left\{\mathbf{q}_{i^*}(\mathbf{x}, \mathbf{W}) - \mathbf{q}_{i^*}^2(\mathbf{x}, \mathbf{W})\right\}\left\{\sum_{j=1}^N \mathbf{q}_j(\mathbf{x}, \mathbf{W})(\mathbf{x}_{i^*}^\top \mathbf{x}_{N+1} - \mathbf{x}_j^\top \mathbf{x}_{N+1})\right\}\right]$$

$$= \mathbb{E}\left[\left(1 - \mathbf{q}_{i^*}(\mathbf{x}, \mathbf{W}^k)\right)\sum_{j=1}^N \frac{\mathbf{q}_j(\mathbf{x}, \mathbf{W}^k)}{\mathbf{q}_{i^*}(\mathbf{x}, \mathbf{W}^k)}(\mathbf{x}_{i^*}^\top \mathbf{x}_{N+1} - \mathbf{x}_j^\top \mathbf{x}_{N+1})\right]$$

$$\leq \mathbb{E}\left[\min\{1, \zeta_k\}\sum_{j=1}^N \exp\left(-\xi_1^k\{\mathbf{x}_{i^*}^\top \mathbf{x}_{N+1} - \mathbf{x}_j^\top \mathbf{x}_{N+1}\}\right)\{\mathbf{x}_{i^*}^\top \mathbf{x}_{N+1} - \mathbf{x}_j^\top \mathbf{x}_{N+1}\}\right]$$

$$\leq N \exp\left(\mathbb{E}[-\xi_1^k\{\mathbf{x}_{i^*}^\top \mathbf{x}_{N+1} - \mathbf{x}_j^\top \mathbf{x}_{N+1}\}]\right) \mathbb{E}[\mathbf{x}_{i^*}^\top \mathbf{x}_{N+1} - \mathbf{x}_j^\top \mathbf{x}_{N+1}]$$

$$\leq N \exp(-\xi_1^k/2)$$

where we define a random variable $\zeta_k$ by

$$\zeta_k = (N-1)\exp\left(\xi_1^k \cdot \{\mathbf{x}_{(2)}^\top \mathbf{x}_{N+1} - \mathbf{x}_{i^*}^\top \mathbf{x}_{N+1}\}\right) + \exp(\xi_1^k \cdot \{1 - \mathbf{x}_{i^*}^\top \mathbf{x}_{N+1}\} - \xi_2^k),$$

the second inequality comes from

$$1 - \mathbf{q}_{i^*}(\mathbf{x}, \mathbf{W}^k) \leq 1 - \frac{\exp(\xi_1^k \cdot \mathbf{x}_{i^*}^\top \mathbf{x}_{N+1})}{\exp(\xi_1^k - \xi_2^k) + (N-1)\exp(\xi_1^k \cdot \mathbf{x}_{(2)}^\top \mathbf{x}_{N+1}) + \exp(\xi_1^k \cdot \mathbf{x}_{i^*}^\top \mathbf{x}_{N+1})}$$
$$\leq (N-1)\exp\left(\xi_1^k \cdot \{\mathbf{x}_{(2)}^\top \mathbf{x}_{N+1} - \mathbf{x}_{i^*}^\top \mathbf{x}_{N+1}\}\right) + \exp(\xi_1^k \cdot \{1 - \mathbf{x}_{i^*}^\top \mathbf{x}_{N+1}\} - \xi_2^k)$$

and the third inequality comes from Jensen's inequality. Combining We now aim to prove the second inequality. Recall that by Eq. (C.11), we have

$$\frac{1}{\eta}(\xi_2^{k+1} - \xi_2^k) = \mathbb{E}[\mathbf{q}_{N+1}(\mathbf{x}, \mathbf{W}^k)\{\mathbf{q}_{i^*}(\mathbf{x}, \mathbf{W}^k) - \sum_{j=1}^N \mathbf{q}_j^2(\mathbf{x}, \mathbf{W}^k)\}],$$

therefore we have

$$\frac{1}{\eta}(\xi_2^{k+1} - \xi_2^k) \leq \mathbb{E}[\mathbf{q}_{N+1}(\mathbf{x}, \mathbf{W}^k)\{\mathbf{q}_{i^*}(\mathbf{x}, \mathbf{W}^k) - \mathbf{q}_{i^*}^2(\mathbf{x}, \mathbf{W}^k)\}]$$

$$\leq \mathbb{E}\left[\min\{1, \zeta_k\}\frac{\mathbf{q}_{N+1}(\mathbf{x}, \mathbf{W}^k)}{\mathbf{q}_{i^*}(\mathbf{x}, \mathbf{W}^k)}\right]$$

$$\leq \mathbb{E}\left[\frac{1}{1 + \exp\left((\mathbf{x}_{i^*}^\top \mathbf{x}_{N+1} - 1)\xi_1^k + \xi_2^k\right)}\right]$$

$$\leq \mathbb{E}[\exp\left((1 - \mathbf{x}_{i^*}^\top \mathbf{x}_{N+1})\xi_1^k - \xi_2^k\right)]$$

$$\leq \exp(2\xi_1^k - \xi_2^k).$$

Thus we conclude the proof of both inequalities. $\qquad\square$

The next lemma is a direct combination of Lemma 8 and Lemma 9.

**Lemma 10.** *When $\xi_1^k \geq 0$, we have*

$$\frac{d}{\eta}(\xi_1^{k+1} - \xi_1^k) \geq \left(1 - \frac{1}{2^N}\right) C_d \exp(-6\xi_1^k) - 2\exp(2\xi_1^k - 2\xi_2^k)$$

The next lemma provides a lower bound for the update of $\xi_3$. Notably, when $\xi_1^k$ is close to $\xi_2^k$, $\xi_2^k$ increases in a larger scale.

**Lemma 11.** *When $\xi_1^k \geq 0$, we have the following inequality holds:*

$$\frac{1}{\eta}(\xi_2^{k+1} - \xi_2^k) \geq \frac{1}{(N+1)^3 e} \exp\left(2a_{n,d} \cdot \xi_1^k - 2\xi_2^k\right),$$

*where $a_{n,d}$ is a constant only pertains to $n$ and $d$ that is defined in Lemma 9*

*Proof.* With Eq. (C.11), we have

$$
\begin{aligned}
\frac{1}{\eta}(\xi_2^{k+1} - \xi_2^k) &\geq \mathbb{E}[\mathbf{q}_{i^*}(\mathbf{x}, \mathbf{W}^k) \cdot \mathbf{q}_{N+1}^2(\mathbf{x}, \mathbf{W}^k)] \\
&= \mathbb{E}\left[\mathbf{q}_{i^*}^3(\mathbf{x}, \mathbf{W}^k)\left(\frac{\mathbf{q}_{N+1}^2(\mathbf{x}, \mathbf{W}^k)}{\mathbf{q}_{i^*}^2(\mathbf{x}, \mathbf{W}^k)}\right)\right] \\
&= \mathbb{E}\left[\mathbf{q}_{i^*}^3(\mathbf{x}, \mathbf{W}^k)\exp\left(2\xi_1^k(1 - \mathbf{x}_{i^*}^\top \mathbf{x}_{N+1}) - 2\xi_2^k\right)\right] \\
&\geq \frac{1}{e}\mathbb{E}\left[\mathbf{q}_{i^*}^3(\mathbf{x}, \mathbf{W}^k)\exp\left(2a_{n,d} \cdot \xi_1^k - 2\xi_2^k\right)|\mathbf{x}_{i^*}^\top \mathbf{x}_{N+1} \geq a_{n,d}\right] \\
&\geq \frac{1}{(N+1)^3 e}\exp\left(2a_{n,d} \cdot \xi_1^k - 2\xi_2^k\right)
\end{aligned}
$$

where the first inequality comes from Eq. (C.14), second inequality comes from Lemma 19, the third inequality comes from $\xi_1^k \geq 0$. $\square$

**Step (3): Upper Bound for $\xi_1^k/\xi_2^k$.** Now, combining Lemma 9 and 11, we immediately get the following result:

**Lemma 12.** *If $\sigma = \xi_2^0 \geq 3\log(\frac{a_{n,d}}{2N^4 d}) = O\left(\max\{\log(Nd), -\log\left(1 - (N\sqrt{d})^{\frac{1}{d}}\right)\}\right)$, and $\xi_1^k \geq 0$ for all $k \geq 0$, then we will have $\xi_1^k \leq \frac{7}{15}\xi_2^k$.*

*Proof.* By Lemma 9, we know that whenever $\xi_1^k \geq 0$, we have

$$\frac{1}{\eta}(\xi_1^{k+1} - \xi_1^k) \leq \frac{2N}{d}\exp(-\xi_1^k/2) - \frac{a_{n,d}}{dN^3 e}\exp(2(\xi_1^k - \xi_2^k)),$$

therefore, if

$$\frac{2N}{d}\exp(-\xi_1^k/2) < \frac{a_{n,d}}{dN^3 e}\exp(2(\xi_1^k - \xi_2^k)),$$

which is equivalent to

$$2.5\xi_1^k > \log(\frac{a_{n,d}}{2N^4 d}) + \xi_2^k,$$

we will have $\xi_1^{k+1} - \xi_1^k < 0$. By Lemma 11, we have $\xi_2^k \geq \xi_2^0 \geq 3\log(\frac{a_{n,d}}{2N^4 d})$. Therefore $\xi_1^{k+1}$ will not increase as long as

$$\frac{15}{7}\xi_1^k \geq \xi_2^k,$$

finally, recall that

$$a_{n.d} = 1 - \frac{1}{(2Nk_d)^{\frac{2}{d-3}}},$$

and our result follows. $\square$

**Step (4): Scale of $\xi_1^k$ and $\xi_2^k$.** Finally, we conclude the convergence of gradient descent with the following two Lemma:

**Lemma 13.** *With $\sigma$ and $N$ satisfying the condition in Theorem 1, we have*

$$\xi_1^k, \xi_2^k = \Omega(\eta \cdot \mathrm{poly}(N, d) \cdot \log k),$$

*with $\xi_1^k \leq \frac{1}{2}\xi_2^k$ holds for all $k \geq 0$.*

*Proof.* We first establish a convergence rate for $\xi_2$. Note that by Lemma 9, Lemma 11 and Lemma 12, we have

$$\xi_2^{k+1} - \xi_2^k \leq \eta \cdot \exp(-\frac{1}{15}\xi_2^k),$$

and

$$\xi_2^{k+1} - \xi_2^k \geq \eta \cdot \frac{1}{(N + 1^3 e)} \exp(-2\xi_2^k)$$

holds for all $k \geq 0$. Therefore, we have $\xi_2^k = \Omega(\eta \cdot \mathrm{poly}(N, d) \cdot \log k)$ for all $k \geq 0$. Now we turn to $\xi_1^k$. By Lemma 11, when

$$8\xi_1^k \leq \xi_2^k - \log\left(\frac{C_d\left(1 - \frac{1}{2^N}\right)}{4}\right),$$

we have

$$\xi_1^{k+1} - \xi_1^k \geq \frac{\eta}{d}\left(1 - \frac{1}{2^N}\right)C_d \exp(-6\xi_1^k) \geq 0.$$

Since $\xi_2^0 = \sigma \geq 2\log\left(\frac{C_d\left(1 - \frac{1}{2^N}\right)}{4}\right)$, $\xi_2^k$ monotonically increasing, and $\xi_1^1 \geq 0$, by induction, we have $\xi_1^k \geq 0$ for all $k$, and $\xi_1^k \geq O(\mathrm{poly}(N, d)\log k)$ until $\xi_1^k \geq \frac{\xi_2^k}{8}$. Meanwhile, Lemma 12 shows that

$$\xi_1^{k+1} - \xi_1^k \leq \frac{2N}{d}\exp(-\xi_1^k/2)$$

for all $k \geq 0$, which implies $\xi_1^k = O(\mathrm{poly}(N, d)\log k)$. Therefore, $\xi_1^k = \Omega(\mathrm{poly}(N, d)\log k)$. The results of $\xi_1^k \leq \frac{1}{2}\xi_2^k$ follows by Lemma 12 and $\xi_1^k \geq 0$. □

### C.3 Convergece of Loss Function $L(\mathbf{W})$

We also have the following bound for the loss function.

**Lemma 14.** *When $\mathbf{W}$ defined in Eq. (2.4) satisfies $\mathbf{W}_{11} = \xi_1 I_d$, $\mathbf{W}_{33} = -\xi_2$, with $\xi_1 \geq 0$, and the rest of items are all zero matrices, we have*

$$\mathbb{E}\left[\left(\sum_{j=1}^{N+1} \mathbf{q}(\mathbf{x}, \mathbf{W})\mathbf{y}_j - \mathbf{y}_{i*}\right)^2\right] \leq O\left(\frac{N^3 k_d^2}{\xi_1}\right) + \exp(2\xi_1 - \xi_3)$$

*Proof.* We have

$$\mathbb{E}\left[\left(\sum_{j=1}^{N+1} \mathbf{q}(\mathbf{x}, \mathbf{W})\mathbf{y}_j - \mathbf{y}_{i*}\right)^2\right] = 1 + \mathbb{E}\left[\sum_{j=1}^{N} \mathbf{q}_j^2(\mathbf{x}, \mathbf{W})\right] - 2\mathbb{E}\left[\mathbf{q}_{i*}(\mathbf{x}, \mathbf{W})\right]$$

$$= \mathbb{E}[1 - \mathbf{q}_{i*}(\mathbf{x}, \mathbf{W})] + \mathbb{E}\left[\sum_{j=1}^{N} \mathbf{q}_j^2(\mathbf{x}, \mathbf{W}) - \mathbf{q}_{i*}(\mathbf{x}, \mathbf{W})\right],$$

note that

$$\mathbb{E}\left[\sum_{j=1}^{N} \mathbf{q}_j^2(\mathbf{x}, \mathbf{W}) - \mathbf{q}_{i*}(\mathbf{x}, \mathbf{W})\right] \leq \mathbb{E}[\sum_{j=1}^{N} \mathbf{q}_{i*}(\mathbf{x}, \mathbf{W})\mathbf{q}_j(\mathbf{x}, \mathbf{W}) - \mathbf{q}_{i*}(\mathbf{x}, \mathbf{W})]$$

$$\leq \mathbb{E}\left[-\mathbf{q}_{N+1}(\mathbf{W}, \mathbf{x})\mathbf{q}_{i*}(\mathbf{x}, \mathbf{W})\right]$$

$$\leq 0,$$

Therefore $\mathbb{E}\left[\left(\sum_{j=1}^{N+1}\mathbf{q}(\mathbf{x},\mathbf{W})\mathbf{y}_j - \mathbf{y}_{i^*}\right)^2\right] \leq \mathbb{E}[1 - \mathbf{q}_{i^*}(\mathbf{x},\mathbf{W})]$. However, we have

$$
\mathbb{E}\big[1 - \mathbf{q}_{i^*}(\mathbf{x},\mathbf{W})\big] \leq \mathbb{E}\Big[\sum_{j\neq i^*, j\in[N+1]}\mathbf{q}_j(\mathbf{x},\mathbf{W})\Big]
$$

$$
= \mathbb{E}\Big[\mathbf{q}_{i^*}(\mathbf{x},\mathbf{W})\sum_{j\neq i^*, j\in[N+1]}\frac{\mathbf{q}_j(\mathbf{x},\mathbf{W})}{\mathbf{q}_{i^*}(\mathbf{x},\mathbf{W})}\Big]
$$

$$
\leq (N-1)\mathbb{E}\Big[\exp\big(\xi_1\cdot\{\mathbf{x}_j^\top\mathbf{x}_{N+1} - \mathbf{x}_{i^*}^\top\mathbf{x}_{N+1}\}\big)\Big] + \mathbb{E}[\exp(\xi_1(1 - \mathbf{x}_{i^*}^\top\mathbf{x}_{N+1}) - \xi_3)],
$$

$$(\mathbf{q}_{i^*} \leq 1)$$

$$
\leq (N-1)\mathbb{E}\Big[\exp\big(\xi_1\cdot\{\mathbf{x}_j^\top\mathbf{x}_{N+1} - \mathbf{x}_{i^*}^\top\mathbf{x}_{N+1}\}\big)\Big] + \exp(2\xi_1 - \xi_3),
$$

By Lemma 20, we have

$$
\mathbb{E}\left[\left(\sum_{j=1}^{N+1}\mathbf{q}(\mathbf{x},\mathbf{W})\mathbf{y}_j - \mathbf{y}_{i^*}\right)^2\right] \leq O\Big(\frac{N^3 k_d^2}{\xi_1}\Big) + \exp(2\xi_1 - \xi_3),
$$

and we conclude the proof. $\qquad\square$

Now, combine Lemma 15 with our dynamic bounds for $\xi_1^k$ and $\xi_2^k$ developed in Section C.2, we have the following convergence result for the loss function.

**Lemma 15.** *When $\mathbf{W}^K$ defined in Eq. (2.4) is updated by gradient descent in Eq. (2.6), we have*

$$
\mathbb{E}\left[\left(\sum_{j=1}^{N+1}\mathbf{q}(\mathbf{x},\mathbf{W})\mathbf{y}_j - \mathbf{y}_{i^*}\right)^2\right] \leq O\Big(\frac{\mathrm{poly}(N,d)}{\log K}\Big).
$$

*Proof.* By Lemma 15, we have

$$
\mathbb{E}\left[\left(\sum_{j=1}^{N+1}\mathbf{q}(\mathbf{x},\mathbf{W}^k)\mathbf{y}_j - \mathbf{y}_{i^*}\right)^2\right] \leq O\Big(\frac{N^3 k_d^2}{\xi_1^k}\Big) + \exp(2\xi_1^k - \xi_3^k).
$$

By Lemma 13, we have $\xi_1^k \leq \frac{7}{15}\xi_2^k$, with $\xi_1^k = \Omega(\mathrm{poly}(N,d)\log k)$. Thus we have

$$
\mathbb{E}\left[\left(\sum_{j=1}^{N+1}\mathbf{q}(\mathbf{x},\mathbf{W}^k)\mathbf{y}_j - \mathbf{y}_{i^*}\right)^2\right] \leq O\Big(\frac{N^3 k_d^2}{\xi_1^k}\Big) + \exp\big(-\frac{1}{15}\xi_3^k\big)
$$

$$
= \frac{\mathrm{poly}(N,d)}{\log k} + \frac{1}{k^{1/15}}
$$

$$
= \frac{\mathrm{poly}(N,d)}{\log k}.
$$

$\qquad\square$

# D   Proof for Theorem 2 and Corollary 1

In this section, we discuss the behavior of the pretrained transformer in tasks under distribution shift, and provide proof for Theorem 2 and Corollary 1 in Section 3.2.

## D.1 Proof for Theorem 2

In this section, we provide a proof for Theorem 2. The result comes from the following observation. First, we condition our analysis on the $A_\delta$, i.e. assuming that there exists a constant $\delta$ such that

$$\|\mathbf{x}_j - \mathbf{x}_{N+1}\|_2^2 \geq \delta + \|\mathbf{x}_{i^*} - \mathbf{x}_{N+1}\|_2^2, \qquad \forall j \neq i^*, \tag{D.1}$$

Then we have

$$
\begin{aligned}
|\widehat{\mathbf{y}}(\mathbf{W}^k) - \mathbf{y}_{i^*}| &= R \left| \sum_{j=1}^N \mathbf{q}_j(\mathbf{x}, \mathbf{W}^K) \mathbf{y}_j - \mathbf{y}_{i^*} \right| \\
&= \left| \sum_{j \in [N], \mathbf{y}_j \neq \mathbf{y}_{i^*}} \mathbf{q}_j(\mathbf{x}, \mathbf{W}^K)(\mathbf{y}_j - \mathbf{y}_{i^*}) \right| + R|\mathbf{q}_{N+1}(\mathbf{x}, \mathbf{W}^K)| \\
&= 2R \sum_{j \in [N], \mathbf{y}_j \neq \mathbf{y}_{i^*}} \mathbf{q}_j(\mathbf{x}, \mathbf{W}^K) + R \cdot \mathbf{q}_{N+1}(\mathbf{x}, \mathbf{W}^K) \\
&\leq 2R \sum_{j \in [N], \mathbf{y}_j \neq \mathbf{y}_{i^*}} \frac{\mathbf{q}_j(\mathbf{x}, \mathbf{W}^K)}{\mathbf{q}_{i^*}(\mathbf{x}, \mathbf{W}^K)} + R \cdot \frac{\mathbf{q}_{N+1}(\mathbf{x}, \mathbf{W}^K)}{\mathbf{q}_{i^*}(\mathbf{x}, \mathbf{W}^K)} \\
&= 2R \sum_{j \in [N], \mathbf{y}_j \neq \mathbf{y}_{i^*}} \exp\left(\xi_1^K \{\mathbf{x}_j^\top \mathbf{x}_{N+1} - \mathbf{x}_{i^*}^\top \mathbf{x}_{N+1}\}\right) + R \cdot \exp\left(\xi_1^K(1 - \mathbf{x}_{i^*}^\top \mathbf{x}_{N+1}) - \xi_3^K)\right) \\
&\leq 2R \cdot N \exp(-\xi_1^K \cdot \delta) + R \cdot \exp\left(-\frac{1}{15}\xi_3^k\right) \\
&= 2RN \exp\left(-\mathrm{poly}(N, d) \cdot \delta \log K\right) \\
&= O\left(RNK^{-\mathrm{poly}(N,d)\delta}\right)
\end{aligned}
\tag{D.2}
$$

uniformly for all $\{(\mathbf{x}_i, \mathbf{y}_i)\}_{i \in [N]}$ and $\mathbf{x}_{N+1}$ on $A_\delta$. Here the second inequality comes from Theorem 1, in which we show that $\xi_1^K \leq \frac{7}{15}\xi_2^K$. Recall that we defined $\mathbf{q}_j(\mathbf{x}, \mathbf{W}^k)$ in Eq. (C.2) as the weight for the $j$-th label in $\{\mathbf{y}_j\}_{j \in [N+1]}$. Next, we have

$$
\begin{aligned}
\mathbb{E}_{\mathbb{P}^{\mathrm{test}}}\left[\left(\widehat{\mathbf{y}}(\mathbf{W}^k) - \mathbf{y}_{i^*}\right)^2\right] &= \mathbb{E}_{\mathbb{P}^{\mathrm{test}}}\left[\left(\widehat{\mathbf{y}}(\mathbf{W}^k) - \mathbf{y}_{i^*}\right)^2 \mathbb{1}_{A_\delta}\right] + \mathbb{E}_{\mathbb{P}^{\mathrm{test}}}\left[\left(\widehat{\mathbf{y}}(\mathbf{W}^k) - \mathbf{y}_{i^*}\right)^2 \mathbb{1}_{A_\delta^c}\right] \\
&\leq O\left(R^2 N^2 K^{-\mathrm{poly}(N,d)\delta}\right) + 4R^2 \cdot \mathbb{P}^{\mathrm{test}}(A_\delta^c),
\end{aligned}
\tag{D.3}
$$

where the inequality comes from Eq. (D.2). Here the expectation is taken over the testing distribution $\mathbb{P}^{\mathrm{test}}$. By taking inferior on the inequality above for all $\delta > 0$, we conclude our proof for Theorem 2.

## D.2 Proof for Corollary 1

In this section, we provide the proof for Corollary 1. The first statement of Corollary 1 comes from Markov's inequality:

$$
\begin{aligned}
\mathbb{P}^{\mathrm{test}}\left(\mathrm{Round}\left(\widehat{\mathbf{y}}_{\mathbf{W}}(\mathbf{x}_{N+1})\right) \neq \mathbf{y}_{i^*}\right) &= \mathbb{P}^{\mathrm{test}}\left(|\widehat{\mathbf{y}}_{\mathbf{W}}(\mathbf{x}_{N+1}) - \mathbf{y}_{i^*}| \geq \frac{1}{2}\right) \\
&\leq 4 \cdot \mathbb{E}[|\widehat{\mathbf{y}}_{\mathbf{W}}(\mathbf{x}_{N+1}) - \mathbf{y}_{i^*}|^2] \\
&\leq O\left(MN \cdot \inf_{\delta \geq 0}\left\{K^{-\mathrm{poly}(N,d)\delta} + \mathbb{P}^{\mathrm{test}}(A_\delta^c)\right\}\right),
\end{aligned}
$$

where the last inequality comes from Theorem 2. Next, we prove the second statement. Suppose $\mathbb{P}^{\mathrm{test}}(A_{\delta^*}) = 1$ for some $\delta^* > 0$. Then similar to the argument in Eq. (D.2), we have

$$|\widehat{\mathbf{y}}(\mathbf{W}^k) - \mathbf{y}_{i^*}| \leq O\left(MNK^{-\mathrm{poly}(N,d)\delta^*}\right)$$

holds for all $\{(\mathbf{x}_i, \mathbf{y}_i)\} \cup \{\mathbf{x}_{N+1}\} \sim \mathbb{P}^{\text{test}}$ almost surely. Note that

$$\text{Round}\left(\widehat{\mathbf{y}}_{\mathbf{W}}(\mathbf{x}_{N+1})\right) = \mathbf{y}_{i^*}$$

holds whenever

$$|\widehat{\mathbf{y}}(\mathbf{W}^k) - \mathbf{y}_{i^*}| < \frac{1}{2},$$

therefore, it suffices to have $\frac{1}{2} \leq O\left(MNK^{-\text{poly}(N,d)\delta^*}\right)$, which is equivalent to

$$K = O\left(\frac{\log(MN)}{\text{poly}(N,d)\delta^*}\right).$$

# E   Nonconvexity of Loss Function

In this section, we show that the loss function is defined by Eq. (2.5). We prove by a special subspace of $\mathbf{W} \in \mathbb{R}^{(d+2)\times(d+2)}$ defined by two scalars $\xi_1, \xi_2$,

$$\mathbf{W} = \text{diag}\{\underbrace{\xi_1, \ldots, \xi_1}_{d \text{ times}}, 0, \xi_2\}. \tag{E.1}$$

By showing $L(\mathbf{W})$ is nonconvex under such parametrization, we conclude our proof.

**Lemma 16** (Nonconvexity of Transformer Optimization). *When $\mathbf{W}$ is a two-dimensional subspace of $\mathbb{R}^{(d+2)\times(d+2)}$ defined by Eq.(E.1), the original loss function defined in Eq. (2.5) degenerates to the following:*

$$L(\xi_1, \xi_2) := \mathbb{E}\left[\left(\frac{\sum_{j=1}^{N} \exp(\xi_1 \langle \mathbf{x}_j, \mathbf{x}_{N+1}\rangle)\mathbf{y}_j}{\sum_{i=1}^{N} \exp(\xi_1 \langle \mathbf{x}_i, \mathbf{x}_{N+1}\rangle) + \exp(\xi_1 - \xi_2)} - \mathbf{y}_{i^*}\right)^2\right],$$

*where we use $\mathbf{y}_{N+1} = 0$. Such loss function is still, in general, nonconvex.*

*Proof.* The degeneracy of the original Eq. (2.5) to $L(\xi_1, \xi_2)$ can be shown by basic algebra. We only need to show the nonconvexity of $L(\xi_1, \xi_2)$ in our proof. Note that by Assumption 1, the gradient of $L(\xi_1, \xi_2)$ is defined by

$$\begin{aligned}
\partial_{\xi_2} L(0, \xi_2) &= \partial_{\xi_2} \mathbb{E}\left[\left(\frac{\sum_{j=1}^{N} \mathbf{y}_j}{N + \exp(-\xi_2)} - \mathbf{y}_{i^*}\right)^2\right] \\
&= \partial_{\xi_2} \mathbb{E}\left[1 - 2\frac{\sum_{j=1}^{N} \mathbf{y}_j \mathbf{y}_{i^*}}{N + \exp(-\xi_2)} + \frac{(\sum_{j=1}^{N} \mathbf{y}_j)^2}{(N + \exp(-\xi_2))^2}\right] \\
&= \partial_{\xi_2}\left\{\frac{N}{(N + \exp(-\xi_2))^2} - \frac{2}{N + \exp(-\xi_2)}\right\} \\
&= \frac{-\exp(-2\xi_2)}{(N + \exp(-\xi_2))^3},
\end{aligned} \tag{E.2}$$

where the third equation comes from

$$\begin{aligned}
\mathbb{E}\left[\sum_{j=1}^{N} \mathbf{y}_j \mathbf{y}_{i^*}\right] &= \mathbb{E}\left[\mathbb{E}\left[\sum_{j=1}^{N} \mathbf{y}_j \mathbf{y}_{i^*}\bigg|\{\mathbf{x}_i\}_{i\in[N]}\right]\right] \\
&= \mathbb{E}\left[\sum_{j=1}^{N} \mathbb{1}_{j=i^*} \mathbb{E}\left[\mathbf{y}_j \mathbf{y}_{i^*}\bigg|\{\mathbf{x}_i\}_{i\in[N]}\right]\right] \\
&= \mathbb{E}\left[\sum_{j=1}^{N} \mathbb{1}_{j=i^*} \mathbb{E}\left[\mathbf{y}_{i^*}^2\bigg|\{\mathbf{x}_i\}_{i\in[N]}\right]\right] \\
&= 1
\end{aligned}$$

and $\mathbb{E}[(\sum_{j=1}^{N} \mathbf{y}_i)^2] = \mathbb{E}[\sum_{j=1}^{N} \mathbf{y}_i^2] = N$. Eq. (E.2) shows that when $\lim_{\xi_2 \to +\infty} \partial_{\xi_2} L(0, \xi_2) = 0$ and $\lim_{\xi_2 \to -\infty} \partial_{\xi_2} L(0, \xi_2) = 0$. If $L(\xi_1, \xi_2)$ is convex, then $\partial_{\xi_2} L(0, \xi_2)$ is monotonically increasing, which means $\partial_{\xi_2} L(0, \xi_2) = 0$ for all $\xi_2$. However, this is clearly a contradiction. $\qquad\square$

# F  Auxiliary Lemma

**Lemma 17** (Distribution of Sphere Inner Product). *With the assumption of $\mathbf{x}_i$ sampled from a uniform distribution on the sphere, Let $\tau$ be the cosine of the angle between an arbitrary $d$-dimensional vector and a vector chosen uniformly at random from the unit sphere. Then the probability density function of random variable $\tau \in [-1, 1]$ is $f_\tau(t) = \frac{2\Gamma(\frac{d}{2})}{\sqrt{\pi}\Gamma(\frac{d-1}{2})} \cdot (1 - t^2)^{\frac{d-3}{2}}$.*

*Proof.* For convenience, we assume that $\mathbf{x}_{N+1} = \mathbf{e}_1$. Note that this does not change the distribution of $\mathbf{x}_{N+1} \cdot \mathbf{x}_i$ due to rotation invariance. Let $X_i \sim N(0, 1)$, define

$$Y_1 = X_1, \dots Y_{d-1} = X_{d-1}, Y_d = \frac{X_d}{\sqrt{\sum_{i=1}^{d} X_i^2}}$$

Note that $Y$ is distributed the same way as $\mathbf{x}_i$. Calculating the Jacobian:

$$J = \begin{bmatrix} 1 & 0 & \cdots & 0 & 0 \\ 0 & 1 & & \vdots & \vdots \\ \vdots & & \ddots & 0 & 0 \\ 0 & \cdots & 0 & 1 & 0 \\ -\frac{X_1 X_d}{[\sum_{i=1}^{d} X_i^2]^{3/2}} & \cdots & & -\frac{X_1 X_{d-1}}{[\sum_{i=1}^{N} X_i^2]^{3/2}} & \frac{\sum_{i=1}^{d} X_i^2}{[\sum_{i=1}^{d} X_i^2]^{3/2}} \end{bmatrix}$$

Since $J$ is of the form:

$$J = \begin{bmatrix} \mathbf{I} & 0 \\ \mathbf{a} & b \end{bmatrix}$$

the determinant is easily evaluated:

$$|J| = \frac{\sum_{i=1}^{d} X_i^2}{\left[\sum_{i=1}^{d} X_i^2\right]^{3/2}}$$

Now to solve for the distribution of $Y_N$.

$$f_\tau(y) = \int_{-\infty}^{\infty} \cdots \int_{-\infty}^{\infty} \frac{1}{|J|} f_x(\mathbf{x}) d\mathbf{x}$$

where

$$f_x(\mathbf{x}) = \prod_{I=1}^{d} \frac{1}{\sqrt{2\pi}} e^{-\frac{x_i^2}{2}} = \frac{1}{(2\pi)^{d/2}} e^{-\frac{1}{2} \sum_{i=1}^{d} x_i^2}$$

Writing the distribution in terms of $t$ :

$$f_\tau(t) = \int_{-\infty}^{\infty} \cdots \int_{-\infty}^{\infty} \frac{\left[\sum_{i=1}^{d-1} y_i^2 + \frac{Y_d^2}{1-t^2} \sum_{i=1}^{d-1} y_i^2\right]^{3/2}}{\sum_{i=1}^{N-1} y_i^2} \frac{1}{(2\pi)^{d/2}} e^{-\frac{1}{2}\frac{1}{1-t^2}\sum_{i=1}^{d-1} y_i^2} dy_1 \cdots dy_{d-1}$$

$$f_\tau(t) = \int_{-\infty}^{\infty} \cdots \int_{-\infty}^{\infty} \frac{1}{(1-t^2)^{3/2}} \sqrt{\sum_{i=1}^{d-1} y_i^2} \frac{1}{(2\pi)^{d/2}} e^{-\frac{1}{2}\frac{1}{1-t^2}\sum_{i=1}^{d-1} y_i^2} dy_1 \cdots dy_{d-1}$$

$$f_\tau(t) = \frac{1}{(1-t^2)^{3/2}} \frac{1}{(2\pi)^{N/2}} \int_{-\infty}^{\infty} \cdots \int_{-\infty}^{\infty} \sqrt{\sum_{i=1}^{d-1} y_i^2 e^{-\frac{1}{2}\frac{1}{1-t^2}} \sum_{i=1}^{d-1} y_i^2} dy_1 \cdots dy_{d-1}$$

Now we can make a substitution to remove $y_d$ from inside the integral. Let $u_n = (1-t^2)^{-1/2} y_n$, we have

$$f_\tau(t) = \frac{1}{(1-t^2)^{3/2}} \frac{1}{(2\pi)^{d/2}} (1-t^2)^{d/2} \int_{-\infty}^{\infty} \cdots \int_{-\infty}^{\infty} \sqrt{\sum_{i=1}^{d-1} u_i^2 e^{-\frac{1}{2} \sum_{i=1}^{d-1} u_i^2}} du_1 \cdots du_{d-1},$$

and

$$f_\tau(t) = (1-t^2)^{(d-3)/2} (2\pi)^{-d/2} \int_{-\infty}^{\infty} \cdots \int_{-\infty}^{\infty} \sqrt{\sum_{i=1}^{d-1} u_i^2 e^{-\frac{1}{2}\sum_{i=1}^{d-1} u_i^2}} du_1 \cdots du_{d-1}$$

The integral can be seen to be a constant so:

$$f_\tau(t) = k_d \left(1-t^2\right)^{\frac{d-3}{2}}$$

or for notational convenience:

$$f_\tau(t) = k_d \left(1-t^2\right)^{\frac{d-3}{2}}$$

Since $f_\tau(y)$ is a PDF and is defined for $0 \le t \le 1$ :

$$\int_0^1 k_d \left(1-t^2\right)^{\frac{d-3}{2}} dt = 1, \text{ where } k_d = \frac{1}{\int_0^1 (1-y^2)^{\frac{d-3}{2}} dy}.$$

Furthermore, for $d > 1$, we have $k_d = \frac{2\Gamma\left(\frac{d}{2}\right)}{\sqrt{\pi}\Gamma\left(\frac{d-1}{2}\right)}$. $\qquad\square$

**Lemma 18.** *With $N = O\left(\log d \cdot \sqrt{d}\right)$, we have $\mathbb{E}[\mathbf{x}_{i*}^\top \mathbf{x}_{N+1}] \ge \frac{2}{(N+1)^2}$.*

*Proof.* Note that we have

$$\mathbb{E}[\mathbf{x}_{i*}^\top \mathbf{x}_{N+1}] \ge \mathbb{P}\left(\max_{i\in[N]} \mathbf{x}_i^\top \mathbf{x}_{N+1} \ge \alpha\right) \cdot \alpha + \left\{1 - \mathbb{P}\left(\max_{i\in[N]} \mathbf{x}_i^\top \mathbf{x}_{N+1} \ge \alpha\right)\right\} \cdot (-1).$$

To ensure that $\mathbb{E}[\mathbf{x}_{i*}^\top \mathbf{x}_{N+1}] \ge \frac{2}{(N+1)^2}$, we only need

$$\alpha\mathbb{P}\left(\max_{i\in[N]} \mathbf{x}_i^\top \mathbf{x}_{N+1} \ge \alpha\right) - \left\{1 - \mathbb{P}\left(\max_{i\in[N]} \mathbf{x}_i^\top \mathbf{x}_{N+1} \ge \alpha\right)\right\} \ge \frac{2}{(N+1)^2},$$

which is equivalent to

$$\mathbb{P}\left(\max_{i\in[N]} \mathbf{x}_i^\top \mathbf{x}_{N+1} \ge \alpha\right) \ge \frac{1}{1+\alpha}\left(1 + \frac{2}{(N+1)^2}\right).$$

By Lemma 17, we have

$$\mathbb{P}\big(\max_{i\in[N]} \mathbf{x}_i^\top \mathbf{x}_{N+1} \geq \alpha\big) = 1 - \left(k_d \int_{-1}^{\alpha} (1-t^2)^{\frac{d-3}{2}} \mathrm{d}t\right)^N.$$

Therefore, we only need

$$1 - \left(k_d \int_{-1}^{\alpha} (1-t^2)^{\frac{d-3}{2}} \mathrm{d}t\right)^N = \mathbb{P}\big(\max_{i\in[N]} \mathbf{x}_i^\top \mathbf{x}_{N+1} \geq \alpha\big) \geq \left(1 + \frac{2}{(N+1)^2}\right) \cdot \frac{1}{1+\alpha},$$

which suffices by

$$N \geq \frac{\log\left(1 - (1 + \frac{2}{(N+1)^2}) \cdot \frac{1}{1+\alpha}\right)}{\log(1 - k_d \int_{\alpha}^{1} (1-t^2)^{\frac{d-3}{2}} \mathrm{d}t)}.$$

Note that

$$\int_{\alpha}^{1} (1-t^2)^{\frac{d-3}{2}} \mathrm{d}t \geq \int_{\alpha}^{1} (1-t)^{d-3} \mathrm{d}t = \frac{1}{d-2}(1-\alpha)^{d-2},$$

therefore we only need

$$N \geq \frac{\log(\frac{\alpha}{1+\alpha} - \frac{\alpha}{(1+\alpha)(N+1)^2})}{\frac{k_d}{d-2}(1-\alpha)^{d-2}}.$$

Now, let $\alpha = \frac{1}{d-2}$, since $k_d = \frac{\Gamma(\frac{d}{2})}{\Gamma(\frac{d-1}{2})} = O(\sqrt{d})$, we only need $N \geq O(\frac{d}{e\sqrt{d}}\log(d-2)) = O(\sqrt{d}\log d)$. $\qquad\square$

**Lemma 19** (Concentration upper bound for $\mathbf{x}_{i*}^\top \mathbf{x}_{N+1}$). *When $\mathbf{x}_{i*}^\top \mathbf{x}_{N+1}$ is defined by $\max_{i\in[N]}\{\mathbf{x}_i^\top \mathbf{x}_{N+1}\}$, where $\mathbf{x}_i$ are independently sampled from a uniform sphere distribution, we have*

$$\mathbb{P}\left(\mathbf{x}_{i*}^\top \mathbf{x}_{N+1} \leq 1 - \frac{1}{(2Nk_d)^{\frac{2}{d-3}}}\right) \geq \frac{1}{e}.$$

*Proof.* By Lemma 17,

$$\mathbb{P}\left(\mathbf{x}_{i*}^\top \mathbf{x}_{N+1} \leq \alpha\right) = \left(1 - k_d \int_{\alpha}^{1} (1-t^2)^{\frac{d-3}{2}} \mathrm{d}t\right)^N.$$

Note that

$$\int_{\alpha}^{1} (1-t^2)^{\frac{d-3}{2}} \mathrm{d}t \leq (1-\alpha)(1-\alpha^2)^{\frac{d-3}{2}},$$

since $(1 - \frac{1}{N})^N$ is monotonically increasing, we only need $k_d(1-\alpha)(1-\alpha)^{\frac{d-3}{2}} \leq \frac{1}{N}$. Setting

$$\alpha = 1 - \frac{1}{(2Nk_d)^{\frac{2}{d-3}}} \leq \left(1 - \frac{1}{(Nk_d)^{\frac{2}{d-3}}}\right)^{1/2}$$

suffices. $\qquad\square$

**Lemma 20.** *Suppose $\{\mathbf{x}_i\}_{i\in[N+1]}$ are i.i.d. samples from a uniform distribution on a sphere in $\mathbb{R}^d$, with $\mathbf{x}_{i*}^\top \mathbf{x}_{N+1}$ and $\mathbf{x}_{(2)}^\top \mathbf{x}_{N+1}$ being the largest and second largest order statistics among $\{\mathbf{x}_i^\top \mathbf{x}_{N+1}\}_{i\in[N]}$, respectively. Then we have*

$$\mathbb{E}\left[\exp\big(\xi\big(\mathbf{x}_{(2)}^\top \mathbf{x}_{N+1} - \mathbf{x}_{i*}^\top \mathbf{x}_{N+1}\big)\big)\right] \leq O\left(\frac{N^2 k_d^2}{\xi}\right)$$

*Moreover, we have*

$$\mathbb{E}\left[\exp\big(\xi\big(\mathbf{x}_{(2)}^\top \mathbf{x}_{N+1} - \mathbf{x}_{i*}^\top \mathbf{x}_{N+1}\big)\big)\right] = \Omega\left(\frac{1}{\xi}\right),$$

*where $\Omega(\cdot)$ hides constant depends on $N$ and $d$.*

*Proof.* We denote $\mathbf{x}_i^\top \mathbf{x}_{N+1}$ by $Y_i$, $\mathbf{x}_{i*}^\top \mathbf{x}_{N+1}$ by $Y_{i*}$ and $\mathbf{x}_{(2)}^\top \mathbf{x}_{N+1}$ by $Y_{(2)}$. By Lemma 17, we have the density function of $Y_i$ being $f_d(t) = k_d \cdot (1 - t^2)^{\frac{d-3}{2}}$, where $k_d = \frac{2\Gamma(\frac{d}{2})}{\sqrt{\pi}\Gamma(\frac{d-1}{2})}$ . Then by the joint distribution of order statistics [David and Nagaraja, 2004], the joint density function of $Y_{i*}$ and $Y_{(2)}$ is

$$f_{Y_{i*}, Y_2}(t)(y_1, y_2) = N(N-1) \cdot f_d(y_2)f_d(y_1) \cdot F_d^{N-2}(y_2)$$

for $-1 \leq y_2 \leq y_1 \leq 1$. Therefore, we have

$$\mathbb{E}\left[\exp\left(\xi(\mathbf{x}_{(2)}^\top \mathbf{x}_{N+1} - \mathbf{x}_{i*}^\top \mathbf{x}_{N+1})\right)\right] \leq N(N-1)k_d^2 \int_{-1}^1 \int_{-1}^{y_1} (1-y_2^2)^{\frac{d-3}{2}} (1-y_1^2)^{\frac{d-3}{2}} \exp\left(\xi(y_2-y_1)\right) \mathrm{d}y_2 \mathrm{d}y_1$$

$$\leq \frac{N(N-1)}{2} k_d^2 \int_{-1}^1 \int_{-1}^{y_1} (2-y_1^2-y_2^2) \exp\left(\xi(y_2-y_1)\right) \mathrm{d}y_2 \mathrm{d}y_1$$

$$\leq N(N-1)k_d^2 \int_{-1}^1 \int_{-1-y_1}^0 \exp(\xi y_2) \mathrm{d}y_2 \mathrm{d}y_1$$

$$\leq N^2 k_d^2 2 \int_{-2}^0 \exp(\xi y_2) \mathrm{d}y_2$$

$$= O\left(\frac{N^2 k_d^2}{\xi}\right),$$

Thus we obtain the upper bound. Next, we establish a lower bound.

$$\mathbb{E}\left[\exp\left(\xi(\mathbf{x}_{(2)}^\top \mathbf{x}_{N+1} - \mathbf{x}_{i*}^\top \mathbf{x}_{N+1})\right)\right] \geq \frac{N(N-1)}{2^N} k_d^2 \int_0^1 (1-y_1^2)^{d-3} \int_0^{y_1} \exp\left(\xi(y_2-y_1)\right) \mathrm{d}y_2 \mathrm{d}y_1$$

$$= \frac{N(N-1)}{2^N \xi} k_d^2 \int_0^1 (1-y_1^2)^{d-3} \left(1 - \exp(-\xi y_1)\right) \mathrm{d}y_1$$

$$\geq \frac{N(N-1)}{e^5 2^N \xi} k_d^2 \int_{1/\sqrt{d}}^{2/\sqrt{d}} (1 - \exp(-\xi/\sqrt{d})) \mathrm{d}y_1$$

$$\geq \frac{N(N-1)}{e^5 2^N \sqrt{d}\xi} k_d^2 (1 - \exp(-\xi/\sqrt{d}))$$

$\square$

