# OpenReview forum: "One-Layer Transformer Provably Learns One-Nearest Neighbor In Context"
_NeurIPS.cc/2024/Conference — NeurIPS 2024 poster_

### Official Review · Reviewer_FpKW · 2024-06-19

**Soundness:** 3
**Presentation:** 2
**Contribution:** 2
**Rating:** 4
**Confidence:** 4

**Summary:**

This paper studies the gradient descent dynamics of a softmax-activated self-attention unit trained on a population loss over in-context learning (ICL) tasks. Each task entails predicting the binary label of a query input, where the true label is the label of the 1-nearest neighbor (1-NN) in the context, and the context and query come from a particular distribution. The main result is that starting from a particular initialization, the key-times-query weight matrix converges to an infinite-valued matrix that exactly implements a 1-NN predictor. An additional result bounds the error of this predictor when the context and query come from any distribution satisfying a weaker assumption than the training distribution, and the query label is again the 1-NN. Empirical results verify these theoretical results in the analyzed setting, with slight modifications.

**Strengths:**

- The results develop understanding of three key areas that are understudied in the in-context learning (ICL) theory literature: (1) the behavior of *softmax*-activated attention (rather than linear attention), (2) ICL of tasks other than linear regression, and (3) gradient-based optimization dynamics.

- Aside from typos, the results are rigorous, well-formulated and non-trivial.

- Regarding non-triviality: Lemma 3 is especially helpful to show that the population loss is still nonconvex even when reduced to the loss over the two scalars.

- The proof sketch is mostly well-written (see below).

- The experiments are well-explained and suggest that some of the simplifications made in the analysis (specific initialization, full-gradient descent on population loss) do not fundamentally change the results.

**Weaknesses:**

1. The required conditions on the training data distribution (context inputs that are uniform on the sphere with labels that are independent Bernoulli random variables with parameter exactly 0.5 and query label that is generated exactly by a 1-NN classifier), especially the training label distribution, are very specific. It is not clear whether the message of the paper can generalize beyond this specific training distribution. Ideally, the paper would present convergence results for a more general class of training distributions, in the same vein as the class of test distributions it considers, and perhaps even when the query label is not generated by a 1-NN classifier. An even more general result may be that the attention unit behaves as a $k$-NN predictor where $k$ depends on some property of the training distribution.

2. The training data distribution is not only specific, but also entails that the label of the query is not generated in the same way as the label of the context examples, which is inconsistent with practice. Specifically each example label must be independent of the corresponding input (as well as all other terms) while the query label does depend on the query and the context examples. The reasoning behind the statement that independence of $\{\mathbf{x}\_{i}\}\_{i\in[N+1]}$ and  $\{\mathbf{y}\_{i}\}\_{i\in[N]}$ “is also essential to properly study in-context learning of one-nearest neighbor” is incorrect; $\mathbf{y}\_i$ can depend on $\mathbf{x}\_i$ and predictors of both forms mentioned will achieve 50% accuracy.

3. The distribution shift result is solid but not surprising since 1-NN should behave similarly on all distributions for which the true label of the query is in fact the label of the closest nearest neighbor in the context — it requires no knowledge of the data to achieve strong performance.

4. From Lemma 4 and 5, it is not clear how $\xi^k_1$ grows slower than $\xi^k_2$, as is claimed and as is needed for the final result. For any $\xi^k\_1,\xi\_2^k$, the upper bound on $\xi\_1^{k+1}-\xi^k\_1$ can be dominated by $\exp( \text{poly}( N, d ) \xi\_1^k)$, which can be much larger than the lower bound on $\xi\_2^{k+1}-\xi^k\_2$ of $\exp( -\text{poly}( N, d ) \xi\_2^k)$ even when $\xi_1^k = \Omega(\xi_2^k)$.

Minor

- The ICL tasks are binary classifications but the loss is the squared error.

- In Theorem 1, the term $\log(1 - (N\sqrt{d}^{1/d}))$ should be $\log(1 - (N\sqrt{d}^{-1/d}))$?

- Equation 2.6 describes gradient ascent, not gradient descent, and the step size is inverted

- Lemma 4 clearly has multiple typos, one of which makes it not clear how to deduce the true statement.

- The term “epoch” is improperly used in the experiments section.

- The caption of Figure 1 says that error bars are plotted but this is false.

**Questions:**

N/A

**Limitations:**

The authors have addressed limitations.

---

> ### Author Rebuttal · Authors · 2024-08-07
>
> >**Q1**: Can the message of the paper be generalized beyond this specific training distribution or even kNN?
>
> **A1**: We confirm that the data lies on a uniform hypersphere is a key assumption for our analysis, as it allows us to transform a metric comparison problem to an inner-production comparison problem, the latter need not take the $\ell^2$-norm of the tokens into consideration. This further enables a single softmax attention layer to approximate 1-NN algorithm, since softmax layer can naturally approximate an argmax operator, which is crucial to our final results. Data distribution beyond the hypersphere distribution could result in large function approximation errors and further hinder learning. As the first paper studying ICL in learning a nonparametric estimator, our motivation in making this assumption is to start from the cleanest and most tractable mathematical framework, while extending our results to more general data distribution is a highly important future direction.
>
> We also agree with the reviewer that learning k-NN is a valuable topic in understanding ICL. However, the extension of our results to k-NN is non-trivial, and cannot be easily achieved with one single attention layer even with multiple heads, as the softmax operator only allows an easy approximation for choosing the closest/most distant $x_i$ in the context with $W_{11}$ goes to $\infty$, but cannot directly approximate token selectors of other orders. We would like to leave the interesting question of learning k-NN with multi-head multi-layer transformers to future works.
>
>
> >**Q2**: The data assumption that the query is not generated in the same way as the label of the context examples is inconsistent with practice; "Independence of $\{x_i\}$
>  and  $\{y_i\}$ is essential to study in-context learning of 1NN” is incorrect: $y_i$ can depend on $x_i$ and predictors of both forms mentioned will achieve 50% accuracy.
>
> **A2**: We agree with the reviewer that $y_i$ can depend on $x_i$ and the predictors
> of the form $\hat{f} = \hat{f}(x_{1},\ldots,x_{N+1})$ or  $\hat{f} = \hat{f}(x_1,y_1,x_2,y_2,\ldots,x_N,y_N)$ can still only achieve $50\%$ prediction accuracy. For example, if the data in the context follows a ground truth linear model, but the linear vector is randomly generated, then these two types of aforementioned predictors can still only achieve $50\%$ prediction accuracy.
> However, we would like to clarify that the goal of this paper is to demonstrate that one-layer transformers can learn one-nearest neighbor in context. If more complicated dependencies exist in the data, it is easy to expect that the transformer model will not learn a clean one-nearest neighbor decision rule. Instead, the transformer may reasonably learn a prediction rule that is some type of “mixture” between in-context linear regression and in-context one-nearest neighbor. Therefore, while studying the case where $x_i$ and $y_i$ are dependent is an interesting future work direction, it may not serve as the best and cleanest example to study the learning of 1-NN decision rule. On the other hand, when assuming $x_i$ and $y_i$ are independent, we can clearly show that the one-layer transformer can cleanly learn to perform 1-NN classification, which motivates us to assume $x_i$ and $y_i$ being independent. Due to the same reason, we are also inclined to assume that the query in the training data also comes from 1-NN label in the context.
>
> >**Q3**: The distribution shift result is solid but not surprising since 1-NN should behave similarly on all distributions for which the true label of the query is in fact the label of the closest nearest neighbor in the context, thus it requires no knowledge of the data to achieve strong performance.
>
> **A3**: We confirm with the reviewer that distribution shift results come from the model “remembering” the 1-NN algorithm. However, our results also surprisingly show that when the $\{x_i\}_{i\in[N]}$ in the testing data are strictly bounded away from the decision boundary, the prediction error could be even better than the training loss. This is also shown in our empirical results in Section 5.
>
> >**Q4**: From Lemma 4 and 5, it is not clear how $\xi_1^k$ grows slower than $\xi_2^k$, as is claimed and as is needed for the final result. For any $\xi_1^k$, $\xi_2^k$, the upper bound on $\xi_1^{k+1} - \xi_1^k$ can be dominated by $\exp⁡(poly(𝑁,𝑑)\xi_1^k)$, which can be much larger than the lower bound on $\xi_2^{k+1} - \xi_2^k$ of $\exp⁡(−poly(𝑁,𝑑)\xi_2^k)$ even when $\xi_1^k=\Omega(\xi_2^k)$.
>
> **A4** We apologize for the typo in Lemma 4. The correct form of Lemma 4 should be $$ \frac{d}{\eta}(\xi_1^{k+1} - \xi_1^k) \leq c_3\cdot \exp\big(-poly(N,d)\cdot \xi_1^k\big) - c_4\cdot \exp\big(2\cdot (\xi_1^k - \xi_2^k)\big),$$ where $poly(N,d)$ is a positive polynomial-order term, as we proved in Lemma 9-10 in appendix. Therefore $\xi_1^k$ will grow slower than $\xi_2^k$, since $\xi_1^{k+1} - \xi_1^k$ would be dominated by $-\exp\big(2\cdot (\xi_1^k - \xi_2^k)\big)$ when $\xi_1^k$ is big and $\xi_1^k$ is close to $\xi_2^k$. We have already revised this error in our current version.
>
> >**Q5**: The ICL tasks are binary classifications but the loss is the squared error.
>
> **A5** Although we choose binary labels in our work for clarity of expression, our results can be easily generated for all distributions with bounded labels. We promise the reviewer we will comment on this in our future version. Meanwhile, even under binary classification, choosing the MSE loss is still beneficial, as it clearly shows that the attention layer can adapt to different ICL tasks under the same objective function.
>
> >**Q6**: Multiple typos.
>
> **A6**: We appreciate the reviewer’s efforts to point these out and have already revised the errors pointed out.

---

> > ### Comment · Reviewer_FpKW · 2024-08-11
> >
> > I thank the authors for their thorough response, and for amending the typo in Lemma 4. I would not be opposed to seeing this paper accepted, since it provides novel, sound and non-trivial results towards addressing an important issue. Also, I am convinced that extensions to input data that is not uniform on the hypersphere and k-NN are highly non-trivial and probably worthy of separate papers. However, my main concern remains in that the results only apply to tasks generated by a very specific 1-NN-based data distribution in which the labels of the context examples are (unrealistically) independent of the input, whereas it is reasonable to expect that one softmax attention unit learns to behave like a 1-NN regressor in other settings as well, e.g. if the tasks are sinusoid regressions with high frequency, the best that softmax attention should be able to do is predict the label of the nearest input example.

---

> > > ### Author Response · Authors · 2024-08-12
> > >
> > > Thank you for your constructive comments, and for clarifying that you are not opposed to accepting our work. We agree that generalizing our current data distribution would greatly strengthen our results. After a careful examination of our proof, we believe that the data assumptions on $y_i$ in Assumption 1 could be extended to the following form:
> > >
> > >
> > > (1) $\mathbb{E}(y_i y_j | \mathbf{x}\_{1:N}] = 0$ and $\mathbb{E}[y_i^2 | \mathbf{x}\_{1:N}] = 1$ for all $i \neq j, i,j \in[N]$.
> > >
> > >
> > > (2) $\mathbb{P}(y\_{1:N} | \mathbf{x}\_{1:N}) = \mathbb{P}(y\_{1:N} | -\mathbf{x}\_{1:N}) $.
> > >
> > >
> > > Note that such an assumption holds for a wide range of data distributions beyond the case where $\mathbf{x}_i$ and $y_i$ are independent. For example, the following data generating process gives $y_i$ that depends on $\mathbf{x}_i$, but the conditions (1) and (2) above still hold:
> > >
> > >
> > > Consider an arbitrary fixed vector $\mathbf{a} \in R^d$ with $\|a\|_2 >2$. Suppose that $\mathbf{x}_i$, $i=1,\ldots,N$ are independently generated from the uniform distribution on the unit sphere, and supposed that given $\mathbf{x}_i$, $y_i$ is generated as follows:
> > >
> > >
> > > - $y_i = 0 $ with probability $1- \frac{1}{ \max \\{\langle \mathbf{a}, \mathbf{x}_i \rangle^2, 1\\}} $,
> > > - $y_i = \max \\{|\langle \mathbf{a}, \mathbf{x}_i \rangle|, 1\\}$ with probability $\frac{1}{2 \max \\{\langle \mathbf{a}, \mathbf{x}_i \rangle^2, 1\\}} $,
> > > - $y_i = -\max \\{|\langle \mathbf{a}, \mathbf{x}_i \rangle|, 1\\}$ with probability $\frac{1}{2 \max \\{\langle \mathbf{a}, \mathbf{x}_i \rangle^2, 1\\}} $.
> > >
> > >
> > > It is easy to verify that $\mathbb{E}[y_i^2|\mathbf{x}\_{1:N}] =  \mathbb{E}[y_i^2|\mathbf{x}_i] = 1$, $\mathbb{E}[y_i y_j|\mathbf{x}\_{1:N}] = \mathbb{E}[y_i y_j|\mathbf{x}\_{i},\mathbf{x}\_{j}] = 0$ and $\mathbb{P}(y\_{1:N} | \mathbf{x}\_{1:N}) = \mathbb{P}(y\_{1:N} | -\mathbf{x}\_{1:N}) $. Moreover, $\mathbf{x}_i$ and $y_i$ are not independent, since $\mathbb{E}[y_i^4| \mathbf{x}_i ] =  \max\\{\langle \mathbf{a}, \mathbf{x}_i \rangle^2, 1 \\}$ is a function of $\mathbf{x}_i$.
> > >
> > >
> > > We will update the paper to include the more general setting under conditions (1),(2) above. We assure that such an extension only requires minor modifications in the paper, and the proofs do not need any significant change. We believe that including such an extension can significantly strengthen our paper, and we hope that it can address your concerns on the limitation of our data models.

---

### Official Review · Reviewer_FWXF · 2024-07-12

**Soundness:** 3
**Presentation:** 3
**Contribution:** 3
**Rating:** 6
**Confidence:** 3

**Summary:**

This paper investigates the ability of single-layer transformers to learn the one-nearest neighbor (1-NN) prediction rule through in-context learning. It focuses on how transformers can handle nonparametric methods like 1-NN classification, moving beyond simpler tasks like linear regression that previous studies have focused on. The main contributions include establishing that a transformer with a softmax attention layer can minimize the training loss to zero in a highly non-convex landscape and behave like a 1-NN predictor under data distribution shifts. The authors establish rigorous theoretical results in line with their investigation.

**Strengths:**

**Originality.** To the best of my knowledge, this paper is the first to theoretically study the behavior of in context learning under distribution shifts for softmax attention transformers. In general, most theoretical works in related areas study either one layer model or linear attention models.

**Quality.** The paper is well written and the claims are well-substantiated with detailed proofs and theorems

**Clarity.** The paper is well-organized and clear. Notations and background are provided well for better understanding. Although I suggest the authors do a grammatical pass since there are minor mistakes throughout the paper.

**Weaknesses:**

1. I like the intuitions and reasoning provided for justifying assumption 1.  However, assuming that the data lies on a hypersphere and assuming no class imbalance seems rather strict to me. Further, it is far from practical. I would like to see more insights with respect to relaxing these assumptions.

2. The results show that the model converges to 1-NN predictor on the training data even under SGD and random initialization. How well does this generalize to relaxing conditions on the input data lying on the hypersphere?

**Questions:**

1. Authors mention in the caption of Figure 5 about “error bars”. However, I don’t see any shaded error curve in their figure.

**Limitations:**

Please refer to the weaknesses section.

---

> ### Author Rebuttal · Authors · 2024-08-07
>
> Thank you for your detailed comments and suggestions. In the following, we will try our best to address your concerns. To accommodate the extensive character count in equations, we will provide our response in multiple parts.
>
>
>
> >**Q1**: Assuming that the data lies on a hypersphere and assuming no class imbalance seems rather strict to me. Further, it is far from practical. What are the insights?
>
> **A1**: Thank you for your advice! We assume no class imbalance and the hypersphere assumption for the training data when studying ICL under 1-NN data setting, as it is the cleanest and the most mathematically tractable setting. Particularly regarding the hypersphere assumption, we consider this setting because it allows us to transform a distance comparison problem to an inner-product comparison problem, making a single self-attention layer capable to compare distances between different tokens. We would also like to point out that this setting could be extended to other data distributions, such as high-dimensional spherical Gaussian distributions, which can be viewed as being almost uniformly distributed on a high-dimensional sphere.
>
> >**Q2**: How well does this generalize to relaxing conditions on the input data lying on the hypersphere under SGD and random initialization?
>
> **A2**: When the input data tokens have significantly different norms, in general, the model could suffer from a large approximation error. This is because the learning of one-nearest neighbor prediction rule by one-layer transformer relies on the fact that when all tokens have the name norm, the a distance comparison problem in one-nearest neighbor is equivalent to an inner-product comparison problem. However, We expect that our result can be extended to settings in which the data points are located around a sphere, such as high-dimensional Gaussian distribution. The same conclusion can also be derived when $W_11$ is randomly initialized with a small variance. Both conclusions can be achieved by utilizing concentration inequalities and a standard perturbation analysis.
>
> >**Q3** Multiple typos throughout the paper.
>
> **A3** We are grateful to the reviewers for pointing this out, and have updated our paper by conducting another grammar check.

---

> > ### Comment · Reviewer_FWXF · 2024-08-10
> >
> > Many thanks for the rebuttal that addresses many of the weaknesses identified and questions raised. I emphasize that all clarifications made during this rebuttal should be made in any revised manuscript to improve clarity of the work. Given my already positive review, I maintain my score.

---

> > > ### Author Response · Authors · 2024-08-12
> > >
> > > Thank you for your positive review. We will carefully revise the paper, and make sure that all clarifications made in the rebuttal are included in the revised version.

---

### Official Review · Reviewer_cVwG · 2024-07-12

**Soundness:** 3
**Presentation:** 3
**Contribution:** 3
**Rating:** 6
**Confidence:** 3

**Summary:**

This submission considers learning to implement 1-NN in-context with a single self-attention layer. In particular, they consider training on in-context learning (ICL) sequences of form $(x1, y1), (x2, y2), … (x_N, y_N), x_{N+1}$, where $x_i$ are sampled from the $d$-dimensional unit sphere independently and with uniform probability, $y_i$ are i.i.d. $\pm 1$ labels independent of $x_i$, and $x_{N+1}$ is a test point with a prediction target equal to the prediction of 1-NN. The authors show that with a specific weight initialization gradient descent on the population loss converges to a global minimum that corresponds to implementing 1-NN in-context. The proof relies on the observation that under the proposed weight initialization most parameters stay zero during optimization and it becomes possible to describe the dynamics with only 2 variables. While the loss function written as a function of these 2 variables is nonconvex, they show that these two variables converge to infinity, with their difference converging to infinity too. This limit corresponds to the 1-NN algorithm.

The authors also prove that the learned algorithm is robust to distribution shifts, with increasing robustness with the number of gradient descent iterations. Finally, they conduct experiments with random weight initialization and show that a single self-attention layer can be trained to implement 1-NN in-context and be robust to distribution shifts.

**Strengths:**

1. Overall the paper is well-written. The related work is properly referenced.
2. Understanding what learning algorithms transformers can implement in-context and what in-context learning algorithms are learned during training is highly important. A large body of work shows that transformers can implement many of the standard supervised learning algorithms. The main findings of this submission are a good contribution to this body of work and show that transformers can implement 1-NN and under certain conditions learn to implement 1-NN when trained on ICL instances.

**Weaknesses:**

My only concern with this submission is the generality of findings.
* While the technique is interesting, it depends critically on the initialization scheme. As the experimental results hint, a single self-attention might be able to learn to implement 1-NN in-context even with standard initialization. It would be great to see a discussion on how the technique employed in this work can be useful for proving convergence under standard initialization.
* As I understand the employed technique is also tied to the *k=1* case of k-NN. It is unclear whether the technique is general enough to be useful for $k>1$.

**Questions:**

* Lines 127-133: I recommend expanding this part a bit. Also, $W_{3,3}$ should be $-\xi_2$ so that the softmax attention peaks on the example with highest dot product (i.e., the closest point as all points are on the unit sphere).
* In the first equation of Lemma 4, the rightmost term should be $\exp(2\xi_1^k - \xi_2^k)$.
* Denoting sequences with index $k$ as $\xi^k_1$ and $\xi^k_2$ is confusing. I recommend using either $\xi^{(k)}_1$ and $\xi^{(k)}_2$ notation or even better, $\xi_k$ and $\zeta_k$.

**Limitations:**

This submission would benefit form a discussion on the generality of the technique (see the "weaknesses" section).

---

> ### Author Rebuttal · Authors · 2024-08-07
>
> Thank you for your detailed comments and suggestions. In the following, we will try our best to address your concerns. To accommodate the extensive character count in equations, we will provide our response in multiple parts.
>
> >**Q1**: It would be great to see a discussion on how the technique employed in this work can be useful for proving convergence under standard initialization.
>
> **A1**: Thanks for bringing this to our attention! When the variance of the random initialization is small enough, with a standard perturbation analysis, by utilizing the Lipschitz condition of the loss function with respect to $W$, we can achieve a similar convergence result with $W_{33}$ being a big negative value. We will add this discussion to our work in the revision.
>
> >**Q2**: As I understand, the employed technique is also tied to the k=1 case of k-NN. It is unclear whether the technique is general enough to be useful for k>1.
>
> **A2**: We confirm with the reviewer that the extension to $k>1$ is nontrivial, and beyond the ability of one layer of attention even with multiple heads. The reason is that one single attention layer suffices to approximate the one-hot vector of the closest/most distant $x_i$ as $W_{11}$ goes to infinity, but cannot directly approximate other $x_i$ in the context. We would like to leave the interesting question of learning k-NN with multi-head multi-layer transformers to future works.
>
> >**Q3**: Minor questions, including typos and writing unclarity
>
> **A3**: We are grateful to the reviewers for pointing out those issues, and will clean up the notations in our future version.

---

> > ### Comment · Reviewer_cVwG · 2024-08-11
> >
> > Thank you for the rebuttal. I will maintain my score.

---

> > > ### Author Response · Authors · 2024-08-12
> > >
> > > Thank you for your positive feedback. In our revision, we will make sure to add clarifications and discussions about the points you have mentioned.

---

### Official Review · Reviewer_cdZv · 2024-07-14

**Soundness:** 3
**Presentation:** 3
**Contribution:** 3
**Rating:** 6
**Confidence:** 2

**Summary:**

This paper studies the theoretical ability of attention layers to implement a 1-nearest-neighbor (1-NN) classifier via in-context learning (ICL). While prior work has studied the ability for transformers to implement algorithms such as linear regression, this paper is the first to establish that attention layers can also implement the 1-NN prediction rule in-context -- a non-parametric learning algorithm, for which in-context learning with attention seems particularly well-suited. The paper contributes the following: a convergence analysis of 1-NN via ICL, a characterization of how ICL 1-NN performs under distribution shift, and under careful initialization, the non-convexity of transformer optimization becomes tractable.

**Strengths:**

- The analysis seems quite complete -- the authors provide theoretical claims surrounding training and initialization, test-time ICL, and out-of-distribution ICL at test time.
- The fact that the convergence analysis is done for 1-NN, a non-parametric learner, is conceptually interesting. To me, this idea makes a lot of sense because attention seems to do some form of non-parametric learning in-context at test time anyway. Overall, this gives me some hope that the analysis could be a useful tool for understanding the in-context learning ability of transformers, more generally.
- The fact that the convergence analysis, which occurs in a non-convex setting, is solvable using more careful analysis is also interesting.

**Weaknesses:**

- It would be great for the authors to contextualize the work a bit more in terms of understanding transformers as language models, more generally. I understand that this work is primarily theoretical, but I feel that it hints at a key point that isn't coming through very strongly in the text: in-context learning with attention, seems to do a form of non-parametric learning at test time. Understanding how attention implements basic non-parametric learning methods such as 1-NN is a great first step toward understanding how attention does this, and it would be nice to include some commentary (or even speculation) on how this paper could fit into this broader narrative.
- Can the analysis be extended in any trivial way to k-NN (e.g. using multiple heads)? There seems to be a lack of commentary on this, and if this is doable in some simple way, this result should be included. If it turns out to be non-trivial, the paper could benefit from commentary on this as well.
- Throughout the paper, the authors refer to the input examples (xs and ys) as being either independent or not independent. It was somewhat unclear to me whether independence was being used throughout, or not.

**Questions:**

- In equation 2.2, the softmax output is directly multiplied with the embedding matrix, meaning that the W_v matrix is set to the identity. While this is spelled out in the text, including this in the equation would improve clarity.
- In Assumption 2, I think $\sigma_1$ should just be $\sigma$.
- Some of the equations run off of the right-hand side of the page in the appendix.

**Limitations:**

As a theoretical work, social impact is non-applicable. However, the authors do not seem to discuss any limitations of their analysis, more broadly.

---

> ### Author Rebuttal · Authors · 2024-08-07
>
> Thank you for your detailed comments and suggestions. In the following, we will try our best to address your concerns. To accommodate the extensive character count in equations, we will provide our response in multiple parts.
>
> >**Q1**: In-context learning with attention, seems to do a form of non-parametric learning at test time. It would be great for the authors to contextualize the work a bit more in terms of understanding transformers as language models, more generally.
>
> **A1**: Thank you for your advice! We confirm with you that our paper follows the framework of a line of theoretical works in studying In-Context Learning ([1][2][3][4]). Our primary aim is to answer the question of **what type of statistical algorithms could an attention layer approximate** under regular first-order optimization methods. A direct application in the language model would be text categorization, in which the input is a sequence of words/phrases and their corresponding labels, while the query is another word/phrase in which the model aims to predict its label. We promise to add this explanation in the introduction of our new version.
>
> >**Q2**: Can the analysis be extended in any trivial way to k-NN (e.g. using multiple heads)?
>
> **A2**: Thanks for bringing this to our attention! We believe the extension of our results to k-NN is non-trivial, and cannot be easily achieved with one single attention layer even with multiple heads, as the softmax operator only allows an easy approximation for choosing the closest/most distant $x_i$ in the context with $W_{11}$ goes to $\infty$, but cannot directly approximate tokens selectors of other orders. We would like to leave the interesting question of learning k-NN with multi-head multi-layer transformers to future works.
>
> >**Q3**: Is the independence between $x_i$ and $y_i$ being used throughout, or not.
>
> **A3**: We make the assumption of $x_i$ and $y_i$ being independent for the purpose of ensuring that the attention layer does gain its prediction power by learning the 1-NN algorithm: the prediction power of the model must come from utilizing the proper comparison between query and context, as any estimator in the form of $\hat{f}(x_{1:N+1})$ or $\hat{f}(x_{1:N}, y_{1:N})$ can achieve at most 50% accuracy.
>
> From a technical perspective, independence condition allows us to achieve a more delicate characterization of gradient descent, which is helpful in achieving the final results.
>
> >**Q4** Minor questions, including typos and writing unclarity
>
> **A4** We highly appreciate your efforts in pointing out those questions and have already revised these issues in our current manuscript.
>
> [1] Training Dynamics of Multi-Head Softmax Attention for In-Context Learning: Emergence, Convergence, and Optimality, Siyu Chen and Heejune Sheen and Tianhao Wang and Zhuoran Yang
>
> [2] In-Context Convergence of Transformers, Yu Huang and Yuan Cheng and Yingbin Liang
>
> [3] Trained Transformers Learn Linear Models In-Context, Ruiqi Zhang and Spencer Frei and Peter L. Bartlett
>
> [4] Transformers as Statisticians: Provable In-Context Learning with In-Context Algorithm Selection, Yu Bai and Fan Chen and Huan Wang and Caiming Xiong and Song Mei

---

### Decision · Program_Chairs · 2024-09-25

**Decision:**

Accept (poster)

**Comment:**

The authors look at the ability of a one-layer softmax transformer to learn one-nearest neighbor following training by GD.  The authors assume the labels are independent of the context and show that GD-trained nets do succeed in learning the 1NN function.

There was a decent level of support for this work among the reviewers, although some expressed reticence. I'm inclined to agree with the positive reviewers, as I think it's quite a nice contribution to know that GD over a transformer can provably learn a 1NN, at least when the labels are independent. Some other works in the DL theory literature also make rather strong assumptions on the generative mechanism (e.g. https://arxiv.org/abs/2305.15141) but I think the key is to provide insight into whether certain learning algorithms are possible to be implemented through training a transformer, and this work succeeds at that.  I recommend acceptance.